# Learning to Make Adherence-Aware Advice

**Guanting Chen**[1], **Xiaocheng Li**[2], **Chunlin Sun**[3], **Hanzhao Wang**[2]

[1] Department of Statistics and Operations Research, UNC-Chapel Hill
[2] Imperial College Business School, Imperial College London
[3] Institute for Computational and Mathematical Engineering, Stanford University
`guanting@unc.edu`
`{xiaocheng.li, h.wang19}@imperial.ac.uk`
`chunlin@stanford.edu`

## Abstract

As artificial intelligence (AI) systems play an increasingly prominent role in human decision-making, challenges surface in the realm of human-AI interactions. One challenge arises from the suboptimal AI policies due to the inadequate consideration of humans disregarding AI recommendations, as well as the need for AI to provide advice selectively when it is most pertinent. This paper presents a sequential decision-making model that (i) takes into account the human's adherence level (the probability that the human follows/rejects machine advice) and (ii) incorporates a defer option so that the machine can temporarily refrain from making advice. We provide learning algorithms that learn the optimal advice policy and make advice only at critical time stamps. Compared to problem-agnostic reinforcement learning algorithms, our specialized learning algorithms not only enjoy better theoretical convergence properties but also show strong empirical performance.

## 1 Introduction

Artificial intelligence (AI) has achieved remarkable success across various aspects of everyday life. However, it is crucial to acknowledge that many of AI's accomplishments have been developed as fully automatic systems (Mnih et al., 2015; Silver et al., 2017). In several important domains like AI-assisted driving (Balachandran et al., 2021) and AI-assisted healthcare (Shaheen, 2021), AI is faced with the challenge of interacting with humans (Mozannar and Sontag, 2020; De et al., 2021), introducing a more intricate and demanding dynamic. This interaction between AI and humans gives rise to two significant issues. Firstly, it is common for humans to reject following AI's advice, and if AI assumes humans' perfect adherence to its advice, the advice generated under this assumption may not be optimal. Secondly, humans may prefer AI to refrain from constant advice-giving, opting for AI intervention only when necessary. They may value their autonomy when performing well but expect AI guidance during critical moments or when they encounter situations in which they are typically less proficient. These considerations underscore the importance of comprehending human behavior and preferences to develop effective and adaptable AI systems for human-AI interactions.

To address the mentioned challenges, in this paper, we provide a decision-making model for human-AI interactions. For the first challenge, the model takes into account the human's **adherence level**, defined as the probability that the human takes the AI's advice. This allows the machine to account for variations in human adherence level when making advice. For the second challenge, the AI model features an action named **defer**, which refrains from giving advice to humans. This feature recognizes that there are instances when humans prefer autonomy and only seek AI guidance during critical moments or situations where they typically struggle. By integrating the adherence level and action deferral into our model, we formulate these challenges as a decision-making problem.

To cater to this specialized decision-making model, we have developed tailored learning algorithms that are both provably convergent and empirically efficient. These algorithms are specifically designed to effectively handle the unique characteristics and challenges of the human-AI interaction setting.

## 1.1 Related Work

**Human-AI interactions**.    Human-AI interactions have long been studied in fields such as robotics. Methods for modeling human behaviors and collaborating with robots (Bobu et al., 2020; Laidlaw and Dragan, 2022; Carroll et al., 2019) have achieved strong empirical performance. Similar to our definition of adherence level, a stream of literature (Chen et al., 2018; Williams et al., 2023) integrates trust (Khavas et al., 2020) as latent factors into the human-AI model and solves Partially Observable Markov Decision Process (POMDP) to get policies with strong empirical outcomes. Our work primarily centers on modeling and establishing theoretical foundations for the human-AI interaction model and the associated learning problems, thereby complementing the existing body of human-AI interaction literature.

**Modeling human-AI interactions**.    On the modeling side, Grand-Clément and Pauphilet (2022) propose the decision-making model that incorporates the adherence level and illustrates that when the adherence level is low, the optimal advice can be different from the optimal decision. Also, see Sun et al. (2022) for an applied setting of interacting with different adherence levels, Shani et al. (2019) for the relationship between the model and the exploration-conscious RL setting, and Jacq et al. (2022) for the so-called lazy-MDP that features an action similar to defer in our setting.

**Machine learning in human-AI interactions**.    Although there has been no literature associated with learning the decision-making model similar to Grand-Clément and Pauphilet (2022) and Jacq et al. (2022), other machine learning approaches have been put forward (Bastani et al., 2021; Meresht et al., 2020; Straitouri et al., 2021; Okati et al., 2021; Chen et al., 2022; Hong et al., 2023; Mao et al., 2023; Mohri et al., 2023) with different human-AI interaction settings.

**Theoretical reinforcement learning**.    Our first proposed algorithm is an optimism-based reinforcement learning method that learns the optimal advice policy. This approach is inspired by the theoretical online reinforcement learning literature (Jaksch et al., 2010; Lattimore and Hutter, 2014; Dann and Brunskill, 2015; Azar et al., 2017; Dann et al., 2017; Zanette and Brunskill, 2019; Domingues et al., 2021). Instead of directly applying the upper confidence bound in the literature, we customize the learning algorithm so that it leverages special properties in our decision-making model, resulting in advantages in theoretical properties and empirical performance. Our second algorithm adopts a reward-free exploration (RFE) approach (Jin et al., 2020), which first explores the environment for a given number of episodes, and then becomes capable of outputting near-optimal policy for any bounded reward functions. We find this approach works well for learning algorithms that make pertinent advice. See Zhang et al. (2020); Kaufmann et al. (2021); Ménard et al. (2021); Miryoosefi and Jin (2022) for the follow-up works in RFE.

Our contribution is twofold:

First, we propose a decision-making model for advice-giving that incorporates human's adherence level and an option for the AI to defer the advice and trust the human. This is a comprehensive modeling framework for effective human-AI interactions, where the optimal decision-making not only considers human adherence level but also makes advice/recommendations only at critical states.

Second, based on this decision-making model, we develop tailored learning algorithms that output near-optimal advice policies and know when to make pertinent advice. Compared to the state-of-the-art problem-agnostic RL algorithms, our algorithm features tighter sample complexity bound and stronger empirical performance.

## 2 Model Setup

Consider a human decision-maker that takes sequential actions under an episodic Markov decision process (MDP) described by the tuple $\mathcal{M}^{\mathtt{H}} = (\mathcal{S}, \mathcal{A}, H, p, r)$. The superscript $\mathtt{H}$ emphasizes the human's involvement in this MDP, $\mathcal{S}$ denotes the set of states, $\mathcal{A}$ denotes the set of actions, $H$ is the horizon of each episode (different from the superscript $\mathtt{H}$), $p$ denotes a deterministic time-dependent transition kernel so that $p_h(s'|s, a)$ is the transition probability from state $s \in \mathcal{S}$ to state $s' \in \mathcal{S}$ under the action $a \in \mathcal{A}$ at time $h$, and $r$ denotes a time-dependent reward function where $r_h(s, a) \in [0, 1]$. Let $S = |\mathcal{S}|$ and $A = |\mathcal{A}|$ denote the cardinality of $\mathcal{S}$ and $\mathcal{A}$, respectively.

Suppose the human follows a fixed (suboptimal) policy $\pi^{\mathrm{H}}$ such that the probability of taking action $a$ at state $s$ and time $h$ is $\pi_h^{\mathrm{H}}(a|s)$. Alongside the human, an intelligent machine makes advice as decision support to improve the reward collected under $\pi_h^{\mathrm{H}}$. In other words, the machine does not seek to change human policy but rather improve its final outcome given its suboptimality. Specifically, upon the arrival at each state, the machine can choose to make advice $a^{\mathrm{M}} \in \mathcal{A}$ to the human (the superscript M stands for the machine), or to trust the human and defer the action to the human, denoted by $a^{\mathrm{M}} = \mathrm{defer}$. If the machine chooses to defer, the human follows its default policy $\pi^{\mathrm{H}}$. If the machine chooses to advise, the human takes the machine's advice with probability $\theta(s, a^{\mathrm{M}}) \in [0,1]$, where $\theta(\cdot, \cdot)$ is the *adherence* level of the human, and is defined as follows.

**Definition 1** *The human's adherence level $\theta : \mathcal{S} \times \mathcal{A} \to [0,1]$ is the probability of human adopting/adhering to the machine's certain advice at a certain state.*

Given the setup, the human takes action $a^{\mathrm{H}}$ according to the following law:

$$
\mathbb{P}_h(a^{\mathrm{H}} = a|s, a^{\mathrm{M}}) = \begin{cases} \pi_h^{\mathrm{H}}(a|s), & \text{if } a^{\mathrm{M}} = \mathrm{defer}, \\ \theta(s, a^{\mathrm{M}}), & \text{if } a^{\mathrm{M}} \neq \mathrm{defer} \text{ and } a = a^{\mathrm{M}} \text{ (adhere)}, \\ (1 - \theta(s, a^{\mathrm{M}})) \cdot \dfrac{\pi_h^{\mathrm{H}}(a|s)}{1 - \pi_h^{\mathrm{H}}(a^{\mathrm{M}}|s)}, & \text{if } a^{\mathrm{M}} \neq \mathrm{defer} \text{ and } a \neq a^{\mathrm{M}} \text{ (not adhere).} \end{cases}
\tag{1}
$$

To summarize, under the human-machine interaction, the underlying dynamic becomes

$$
s_h \xrightarrow{\text{machine makes advice}} a^{\mathrm{M}} \xrightarrow{a^{\mathrm{H}} \sim \mathbb{P}_h(\cdot|s_h, a^{\mathrm{M}})} a^{\mathrm{H}} \xrightarrow{s_{h+1} \sim p_h(\cdot|s_h, a^{\mathrm{H}})} s_{h+1}.
$$

At each time $h$, the machine first makes the advice $a^{\mathrm{M}}$ upon the state $s_h$ and the human incorporates the machine advice into a final action $a^{\mathrm{H}}$, and then transit to the next state $s_{h+1}$.

**The machine's MDP**. From the machine's perspective, the MDP is slightly different from the MDP faced by human. It can be described by $\mathcal{M}^{\mathrm{M}} = (\mathcal{S}, \bar{\mathcal{A}}, H, p^{\mathrm{M}}, r^{\mathrm{M}})$. This MDP shares the same state space $\mathcal{S}$ and horizon $H$ as the human MDP $\mathcal{M}^{\mathrm{H}}$. The action space is augmented to include the defer option $\bar{\mathcal{A}} = \mathcal{A} \cup \{\mathrm{defer}\}$. In the machine's perspective, the transition can be viewed as a direct consequence of making advice $a^{\mathrm{M}} \in \bar{\mathcal{A}}$ (i.e, $s_h \to a^{\mathrm{M}} \to s_{h+1}$), and the transition kernel becomes

$$
p_h^{\mathrm{M}}(s'|s, a^{\mathrm{M}}) = \sum_{a^{\mathrm{H}} \in \mathcal{A}} p_h(s'|s, a^{\mathrm{H}}) \cdot \mathbb{P}_h(a^{\mathrm{H}}|s, a^{\mathrm{M}}),
\tag{2}
$$

where $p_h$ is the transition kernel of the MDP $\mathcal{M}^{\mathrm{H}}$, and the probability $\mathbb{P}_h(\cdot|s, a)$ is specified by the adherence dynamics (1). In parallel, we define the reward by marginalizing human's action

$$
r_h^{\mathrm{M}}(s, a^{\mathrm{M}}) = \sum_{a^{\mathrm{H}} \in \mathcal{A}} r_h(s, a^{\mathrm{H}}) \cdot \mathbb{P}_h(a^{\mathrm{H}}|s, a^{\mathrm{M}}).
$$

Denote $\boldsymbol{\pi} = \{\pi_h\}_{h \in [H]}$ the machine's policy where $\pi_h : \mathcal{S} \to \bar{\mathcal{A}}$. The value function then becomes

$$
V_{h_0}^{\pi}(s) = \mathbb{E}\left[ \sum_{h=h_0}^{H} r_h^{\mathrm{M}}(s_h, a_h) \Big| s_{h_0} = s \right], \quad \text{where } a_h = \pi_h(s_h) \text{ and } s_{h+1} \sim p_h^{\mathrm{M}}(\cdot|s_h, a_h),
$$

and let $V_{H+1}^{\pi}(s) = 0$ for any $s \in \mathcal{S}$. The optimal value $V^*$ and the optimal policy $\pi^*$ are defined by

$$
V_h^*(s) = \max_{\pi \in \Pi} V_h^{\pi}(s), \quad \pi^* = \arg\max_{\pi \in \Pi} V_1^{\pi}(s)
$$

where $\Pi$ consists of all deterministic non-anticipating Markov policies. Similarly, we define the corresponding $Q$ functions to be

$$
Q_h^{\pi}(s, a) = r_h^{\mathrm{M}}(s, a) + \sum_{s' \in \mathcal{S}} p_h^{\mathrm{M}}(s'|s, a) V_{h+1}^{\pi}(s'), \text{ and } Q_h^*(s, a) = r_h^{\mathrm{M}}(s, a) + \sum_{s' \in \mathcal{S}} p_h^{\mathrm{M}}(s'|s, a) V_{h+1}^*(s').
$$

**Human-centric system.** The theme of the formulation and all our following results is a human-centric decision system where the machine acknowledges the suboptimal behavior of the human and makes advice on critical states to improve the reward. So the learning and optimization of our paper take the perspective of the machine (solving $\mathcal{M}^{\mathrm{M}}$) and do not seek to change the underlying human policy $\pi^{\mathrm{H}}$.

## 3 THE LEARNING PROBLEM

Now we discuss learning problems associated with the above human-machine adherence model. We consider two *learning environments* for the problem:

$\mathcal{E}_1$ (Environment 1 – partially known): the environment's state transition kernel $p$, the reward $r$, and the human's behavior policy $\pi^{\text{H}}$, are known; the human's adherence level $\theta$ is unknown.

$\mathcal{E}_2$ (Environment 2 – fully unknown): the environment's state transition kernel $p$, the reward $r$, the human's behavior policy $\pi^{\text{H}}$, and the human's adherence level $\theta$ are unknown.

For $\mathcal{E}_1$, the goal is simply to learn the optimal policy under the unknown adherence level $\theta$. We develop a learning algorithm that outputs $\epsilon$-optimal advice policy and features better sample complexity compared to the vanilla application of problem-agnostic RL methods on $\mathcal{M}^{\text{M}}$. For $\mathcal{E}_2$, we know neither the environment nor the human's policy. Thus the learning problem entails learning the dynamics of both the environment and the human policy. We develop a provably convergent learning algorithm that outputs the optimal policy, and in addition, the learned advice policy only gives advice when necessary (choosing to defer for non-critical steps).

Our investigations on these two learning formulations highlight three points. First, the inherent structure of the human-machine interaction allows more sample-efficient algorithms (than the vanilla application of the off-the-shelf RL algorithms) both theoretically and empirically. Second, the knowledge of the underlying environment ($\mathcal{E}_1$ compared against $\mathcal{E}_2$) significantly, also unsurprisingly, reduces the sample complexity of the learning algorithm. Third, we establish a close connection between the formulation of the human-machine interaction with the problems of reward-free exploration (Jin et al., 2020) and constrained MDPs (Altman, 2021).

### 3.1 MAIN RESULTS

We first state the technical results and then present the detailed algorithms and analyses in the subsequent section.

**Theorem 1 (Environment $\mathcal{E}_1$, informal)** For environment $\mathcal{E}_1$, Algorithm 1 finds an $\epsilon$-optimal advice policy with a PAC sample complexity $O(H^2 S^2 A/\epsilon^2)$ with high probability.

Under environment $\mathcal{E}_1$, Theorem 1 gives a PAC sample complexity for the UCB-type (Upper-Confidence-Bound-type) Algorithm 1. We remark that applying the existing problem-agnostic algorithms can only achieve a suboptimal order of sample complexity on the problem: $O(H^3 S^2 A/\epsilon^2)$ via the model-based algorithm (Dann and Brunskill, 2015) and $O(H^4 SA/\epsilon^2)$ via the model-free algorithm (Jin et al., 2018)[1]. Specifically, the bound in Dann and Brunskill (2015) gives an additional factor of $H$ compared to the bounds in the original setting, where stationary transition density is assumed; this is due to the fact that though the adherence level $\theta$ is stationary, the transition becomes non-stationary when compounding $\theta$ and underlying transition of the human's underlying MDP. Also, we note that such an improvement on $H$ is not due to a reduction in the number of unknown parameters because the adherence level $\theta$ has a dimensionality of $SA$. Indeed, the key to the improvement is the intrinsic structure of the human-machine problem enables a more sample-efficient design of the UCB algorithm (See Section 4.1 for details). Moreover, we also provide another algorithm that finds an $\epsilon$-optimal advice policy with a sample complexity of $O(H^3 SA/\epsilon^2)$ for $\mathcal{E}_2$ (See Algorithm 3 in appendix A.3 for details).

For environment $\mathcal{E}_2$, we assume no prior knowledge at all, and this makes the machine's problem no different than a generic RL problem. Thus we consider a slight twist of the machine's MDP with the notion of *pertinent* advice. This twisted formulation enables richer analytical structures and draws interesting connections with several existing frameworks. Specifically, consider a new machine's MDP $\mathcal{M}_{\beta}^{\text{M}} \in \left( \mathcal{S}, \bar{\mathcal{A}}, H, p^{\text{M}}, r_{\beta}^{\text{M}} \right)$ which inherits everything from $\mathcal{M}^{\text{M}} \in \left( \mathcal{S}, \bar{\mathcal{A}}, H, p^{\text{M}}, r^{\text{M}} \right)$ except for the reward

$$r_{h,\beta}^{\text{M}}(s,a) = r_h^{\text{M}}(s,a) - \beta \cdot \mathbb{I}\{a \neq \text{defer}\}, \tag{3}$$

---

[1]The authors obtain a regret bound instead of PAC sample complexity bound. However, they convert the regret bound to a PAC sample complexity bound in (Jin et al., 2018, Section 3.1)

where the $\mathbb{I}\{\cdot\}$ is the indicator function and $\beta > 0$ is a constant. Under $\mathcal{M}_\beta^\mathtt{M}$, we denote $V_\beta^\pi$ and $V_\beta^*$ the value functions of $\pi$ and the optimal value function, respectively, and the optimal policy $\pi_\beta^* \in \arg\max_\pi V_\beta^\pi$. The new reward function enforces a penalization of $\beta$ for making advice and thus regularizes the number of machine advices throughout the horizon. In practice, providing advice to human at every step can be annoying in applications such as gaming, driving, or sports. Hence, it is crucial to prioritize and selectively deliver advice based on its criticalness – which we term informally as *pertinent* advice. For example, when the human is an expert and already achieves near-optimal performance, there is no need to give advice; also, when the human is under-performing, and the adherence level is low, there is also no need to give advice because it is unlikely to be taken.

**Proposition 1.** *For all $s \in \mathcal{S}$ and $h \in [H]$ such that $\pi_{h,\beta}^*(s) \neq defer$, we have*

$$Q_h^*(s, \pi_{h,\beta}^*(s)) - V_h^{\pi^\mathtt{H}}(s) \geq \beta.$$

The proposition says that if the machine takes $\pi_{h,\beta}^*(s)$ and sticks with the optimal policy afterward, the reward will be at least $\beta$ more than that if the machine chooses to defer all the way till the end. In this light, we can rank the criticalness of making advice at different states by solving $\mathcal{M}_\beta^\mathtt{M}$ with different $\beta$ which gives a better interpretation of this human-machine system.

**Theorem 2 (Environment $\mathcal{E}_2$, informal)** For $\mathcal{E}_2$, Algorithm 2 outputs a family of $\epsilon$-optimal policies $\{\hat{\pi}_\beta\}_{\beta>0}$ for $\{\mathcal{M}_\beta^\mathtt{H}\}_{\beta>0}$ with $O(H^5 SA/\epsilon^2)$ episodes such that the following inequality

$$V_{1,\beta}^*(s_1) - V_{1,\beta}^{\hat{\pi}_\beta}(s_1) \leq \epsilon \tag{4}$$

holds uniformly for all $\beta > 0$ with high probability.

Theorem 2 gives the sample complexity of Algorithm 2 which learns a near-optimal policy for all the models $\{\mathcal{M}_\beta^\mathtt{H}\}_{\beta\geq0}$ simultaneously. Such joint learning not only provides a family of policies for the human to customize $\beta$ according to her/his performance but also gives us a handle to understand which are the critical states where the human's policy can be significantly improved.

# 4 Algorithms and Analyses

In this section, we present the algorithms and analyses that achieve the results mentioned previously.

## 4.1 UCB-based algorithm for $\mathcal{E}_1$

Under $\mathcal{E}_1$, the machine works with a human with unknown adherence level $\theta$. An important property of $\theta$ is as follows. Basically, it states that the team of human and machine achieves a higher optimal reward if the human has a higher adherence level. To emphasize the dependence on $\theta$, we write

$$V_h^\pi(s|\theta) = \mathbb{E}\left[\sum_{h'=h}^H r_{h'}^\mathtt{M}(s_{h'}, a_{h'}) \Big| s_h = s, \text{adherence parameter } \theta\right] \text{ and } V_h^*(s|\theta) = \max_{\pi\in\Pi} V_h^\pi(s|\theta).$$

**Proposition 2** (Monotonicity property). *Suppose $\theta_1 \geq \theta_2$ holds entry-wise, then the following inequality holds for all $s \in \mathcal{S}$ and $h \in [H]$*

$$V_h^*(s|\theta_1) \geq V_h^*(s|\theta_2).$$

Proposition 2 implies that finding an upper bound for the optimal value function reduces to finding an upper bound for $\theta$. Algorithm 1 follows this implication and maintains an optimistic estimate $\bar{\theta}^t$ for the true parameter $\theta$. For each episode, it generates the policy $\hat{\pi}_t$ pretending the $\bar{\theta}^t$ as true, and rolls out the episode according to $\hat{\pi}_t$. Then it updates the estimate with the new observations. The optimistic estimate $\bar{\theta}^t$ takes the form of a standard UCB form with a careful choice of the confidence width and we defer more details to Appendix A.2. The algorithm shares the same intuition as other UCB-based algorithms that, with more and more observations, the confidence bound $\bar{\theta}^t$ will shrink to the true $\theta$, and so does the value functions.

Theorem 1 establishes an $(\epsilon, \delta)$-PAC result for Algorithm 1.

---

**Algorithm 1** UCB-ADherence (UCB-AD)

---

1: Input: Target probability level $\delta$.
2: Initialize $t = 1$, $\mathcal{D}_{t-1} = \emptyset$, and the optimistic estimate $\bar{\theta}^t = \mathbf{1}$.
3: **for** $t = 1, 2, \cdots$ **do**
4:     Solve the advice policy $\hat{\pi}^t = \arg\max_\pi V^\pi(\cdot|\bar{\theta}^t)$ given the current optimistic estimate $\bar{\theta}^t$
5:     Sample a new episode $z_t = \left\{ s_1^t, a_1^{\text{M},t}, a_1^{\text{H},t}, r_1^t, \cdots, s_H^t, a_H^{\text{M},t}, a_H^{\text{H},t}, r_H \right\}$ following policy $\hat{\pi}^t$
6:     Update $\mathcal{D}_t \leftarrow \mathcal{D}_{t-1} \cup \{z_t\}$
7:     Update the optimistic estimate $\bar{\theta}^t \rightarrow \bar{\theta}^{t+1}$ based on $\mathcal{D}_t$ and $\delta$
8: **end for**

---

**Theorem 1.** *For any $\delta \in (0, 1)$, $\epsilon \in (0, 1]$, and $T \in \mathbb{N}^+$, the number of policies among $\{\hat{\pi}^t\}_{t=1}^T$ from Algorithm 1 that are not $\epsilon$-optimal, i.e., $V_1^*(s_1) - V_1^{\hat{\pi}^t}(s_1) > \epsilon$, is bounded by $\tilde{O}\left( \frac{H^2 S^2 A}{\epsilon^2} \cdot \log\frac{1}{\delta} \right)$ with probability $1 - \delta$.*

The proof of the theorem mimics the analysis of Dann and Brunskill (2015). One caveat in the analysis is that the original analysis of Dann and Brunskill (2015) focuses on a stationary setting where transition probabilities depend solely on state and action, remaining independent of the time horizon. However, even when the adherence level $\theta$ remains the same over time, the machine's MDP is non-stationary. An direct adoption is to enlarge the state space to incorporate the horizon step $h$, yet this will result in a sample complexity of $O(H^3 S^2 A / \epsilon^2)$, a worse dependency on $H$. The key is to reduce the upper bound analysis to the adherence level space and utilize Proposition 2 to convert that into a suboptimality gap with respect to the value function. This treatment gives the desirable bound in Theorem 1 which also outperforms the bound from a direct application of results from Azar et al. (2017) to non-stationary MDPs.

## 4.2 REWARD-FREE EXPLORATION ALGORITHM FOR $\mathcal{E}_2$

$\mathcal{E}_2$ has more unknown parameters than $\mathcal{E}_1$ and thus it naturally entails more intense exploration. Moreover, the learning objective becomes more complex: we aim not only to learn the near-optimal policy but also to discern the pertinent advice.

Algorithm 2 is based on the concept of *reward-free exploration* (RFE) (Jin et al., 2020). Specifically, RFE algorithms usually consist of an exploration phase and a planning phase. During the exploration phase, the algorithm collects trajectories from an MDP $\mathcal{M}$ without a pre-specified reward function. In the planning phase, it can compute near-optimal policies of $\mathcal{M}$, given any deterministic reward functions that are bounded.

In our human-machine model, the machine observes $s_h \rightarrow a^{\text{M}} \rightarrow a^{\text{H}} \rightarrow s_{h+1}$, and the trajectory for episode $t$ is $z_t = \{s_1^t, a_1^{\text{M},t}, a_1^{\text{H},t}, r_1^t, s_2^t, a_2^{\text{M},t}, a_2^{\text{H},t}, r_2^t, \cdots, s_H^t, a_H^{\text{M},t}, a_H^{\text{H},t}, r_H^t\}$, where $a_h^{\text{M},t} = \pi^t(s_h^t)$, $a_h^{\text{H},t} \sim \mathbb{P}_h(\cdot|s_h^t, a_h^{\text{M},t})$, and $s_{h+1}^t \sim p_h(\cdot|s_h^t, a_h^{\text{M},t})$. We denote $\hat{p}_h^{\text{M},t}$ and $\hat{r}_h^{\text{M},t}$ the empirical estimation for $p^{\text{M}}$ and $r_h^{\text{M}}$, and $n_h^t(s,a) = \sum_{i=1}^t \mathbb{I}\left\{ \left(s_h^i, a_h^{\text{M},i}\right) = (s,a) \right\}$ the number of times the machine gives advice $a$ at time $h$ and state $s$ in the first $t$ episodes. The key quantity in Algorithm 2 is

$$W_h^t(s,a) = \min\left( H, 16H^2 \frac{\phi(n_h^t(s,a),\delta)}{n_h^t(s,a)} + \left(1 + \frac{1}{H}\right) \sum_{s'} \hat{p}_h^{\text{M},t}(s'|s,a) \max_{a'} W_{h+1}^t(s',a') \right), \tag{5}$$

where $W_{H+1}^t(s,a) = 0$ for $(s,a) \in \mathcal{S} \times \mathcal{A}$, and $\phi(n,\delta)$ grows at the order of $O(\log(n) + \log(1/\delta))$ and is specified in Theorem 2.

Now we formally introduce our Algorithm 2. The algorithm iteratively minimizes an upper bound defined by (5) which measures the uncertainty of a state-action pair, and the upper bound shrinks as the number of visits for the state-action pair increases. The algorithm stops when the upper bound is less than a pre-specified threshold. This algorithm is inspired by the RF-Express algorithm (Ménard et al., 2021), and there is a slight difference in the definition of $W_h^t(s,a)$, $\phi(n,\delta)$ and the stopping rule. In our application, the reward $r^{\text{M}}$ is stochastic and we need to take care of the estimation error; while in Ménard et al. (2021), the algorithm does not need to deal with the reward at all.

---

**Algorithm 2** : RFE-$\beta$

---

1: **Input:** $\epsilon, \delta$, and user-specified $\{\beta_i\}_{i \in \mathcal{I}}$, where $\mathcal{I}$ could be any set where $\beta_i \in [0, H)$
2: **Stage 1: Reward-free exploration**
3: Initialize $t = 1$ and $W_h^t(s, a) = H$ for all $(s, a) \in \mathcal{S} \times \mathcal{A}$
4: Compute $\pi^t$ so that $\pi_h^t(s) = \arg\max_{a \in \mathcal{A}} W_h^t(s, a)$ (see (5))
5: **while** $W_1^t(s_1, \pi^t(s_1)) + 4e\sqrt{W_1^t(s_1, \pi^t(s_1))} > \epsilon/H$ **do**
6:     Sample trajectory $z_t = \{s_1^t, a_1^{\text{M},t}, a_1^{\text{H},t}, r_1^t, \cdots, s_H^t, a_H^{\text{M},t}, a_H^{\text{H},t}, r_H^t\}$ following $\pi^t$
7:     update $t \leftarrow t + 1$, $\mathcal{D} \leftarrow \mathcal{D} \cup \{z_t\}$, $\hat{p}_h^{\text{M},t}(s'|s, a)$, $\hat{r}_h^{\text{M},t}(s, a)$, and $W_h^t(s, a)$
8: **end while**
9: **Stage 2: Policy identification**
10: Use planning algorithms to output optimal advice policy $\{\hat{\pi}_{\beta_i}^\tau\}_{i \in \mathcal{I}}$ for $\left\{ \left( \mathcal{S}, \bar{\mathcal{A}}, H, \hat{p}^{\text{M}}, \hat{r}_{\beta_i}^{\text{M}} \right) \right\}_{i \in \mathcal{I}}$

---

**Theorem 2.** *For $\delta \in (0, 1)$, $\epsilon \in (0, 1]$, and $\phi(n, \delta) = 6 \log(4HSA/(\epsilon\delta)) + S \log(8e(n + 1))$, with probability $1 - \delta$, Stage 1 of Algorithm 2 stops in $\tau$ episodes and*

$$\tau \leq C_1 \frac{H^5 SA}{\epsilon^2} \left( 6 \log(4HSA/(\epsilon\delta)) + S \right),$$

*where $C_1 = \tilde{O}(\log(HSA))$. Moreover, $\{\hat{\pi}_\beta^\tau\}_{\beta > 0}$ have the following property*

$$P \left( V_{1,\beta}^*(s_1) - V_{1,\beta}^{\hat{\pi}_\beta^\tau}(s_1) \leq \epsilon \text{ uniformly for all } \beta \in [0, H) \right) > 1 - \delta.$$

Theorem 2 ensures that Algorithm 2 provides sample estimation for the underlying MDP such that all the policy $\{\hat{\pi}_\beta^\tau\}_{\beta \in [0, H)}$ for pertinent advice are near optimal. The proof is a direct application of the RF-Express (Ménard et al., 2021), except that we have to take care of the estimation error in $\hat{r}^{\text{M}}$. Although Algorithm 2 has the uniform convergence property for any number of bounded reward functions, it can also be used the same way as Algorithm 1, to find the $\epsilon$-optimal policy for $\mathcal{M}^{\text{M}}$ if provided with the non-penalized reward function $\hat{r}^{\text{M}}$. In this context, we can modify RFE-$\beta$ so that with high probability, it solves $\mathcal{M}^{\text{M}}$ with a sample complexity of $O(H^3 SA/\epsilon^2)$ (See Algorithm 3 in Appendix A.3 for details).

**CMDP for pertinent advice.** The algorithm RFE-$\beta$ solves a class of problems $\{\mathcal{M}_\beta^{\text{M}}\}_{\beta > 0}$ simultaneously for all the $\beta$'s and it measures the pertinence of advice by $\beta$. However, sometimes humans lack a quantitative view of how large a $\beta$ value should be considered as pertinent. Here, we introduce a different perspective on how the human should rank the importance of advice, framing it as "in $H$ steps, I want advice no more than $D$ times", and formulate this as a CMDP problem

$$\max_\pi \; \mathbb{E}^\pi \left[ \sum_{h=1}^H r^{\text{M}}(s_h, a_h) \right] \quad s.t. \; \mathbb{E}^\pi \left[ \sum_{h=1}^H \mathbb{I}\{a_h \neq \text{defer}\} \right] \leq D, \tag{6}$$

where $D \in (0, H)$. From the standard primal-dual theorem, this formulation is closely related to the penalty $\beta$ in (3), for the reason that we can treat $\beta$ as a dual variable for the constraint $D$. We refer the reader to the proof of Corollary 1 in Appendix A.3 for details.

Now we present the CMDP method for pertinent advice. After stage 1 of RFE-$\beta$, we solve

$$\max_\pi \; \hat{\mathbb{E}}^\pi \left[ \sum_{h=1}^H \hat{r}^{\text{M},\tau}(s_h, a_h) \right] \quad s.t. \; \hat{\mathbb{E}}^\pi \left[ \sum_{h=1}^H \mathbb{I}\{a_h \neq \text{defer}\} \right] \leq D, \tag{7}$$

where $\hat{\mathbb{E}}$ is the expectation with the underlying transition being $\hat{p}^{\text{M},\tau}$. The next corollary states that $\hat{\pi}_D^\tau$, the solution for (7), is a near-optimal policy for the CMDP (6).

**Corollary 1.** *In the same setting of Theorem 2, for $\delta \in (0, 1)$ and $\epsilon \in (0, 1]$, with probability $1 - \delta$, for all $D \in (0, H)$, $\hat{\pi}_D^\tau$ is a near-optimal solution for the original CMDP (6) such that*

$$V_1^{\hat{\pi}_D^\tau}(s_1) \geq V_1^{\pi_D^*}(s_1) - 2\epsilon, \quad \text{and} \quad \mathbb{E}^{\hat{\pi}_D^\tau} \left[ \sum_{h=1}^H \mathbb{I}\{a_h \neq \text{defer}\} \right] \leq D + \epsilon \tag{8}$$

*where $\pi_D^*$ is the optimal solution for (6).*

Corollary 1 also implies that RFE-$\beta$ can compute near-optimal policies of CMDP (6) for **all** the constraints $D \in [0, H]$, with a sample complexity of $O(H^5 SA/\epsilon^2)$. Compared to other CMDP learning algorithms (for example, $O(H^2 S^3 A/\epsilon^2)$ in Kalagarla et al. (2021)), the sample complexity of Corollary 1 features a lower order in $S$. Moreover, the near-optimal result holds for all constraints $D \in [0, H]$, and for other CMDP learning algorithms, the result only holds for a pre-specified $D$.

## 5 NUMERICAL EXPERIMENT

We perform numerical experiments under two environments: *Flappy Bird* (Williams et al., 2023) and *Car Driving* Meresht et al. (2020). Both Atari game-like environments are suitable and convenient for modeling human behavior while retaining the learning structure for the machine. We focus on the flappy bird environment here and defer the car driving environment to Appendix B.

**Flappy Bird Environment.** We consider a game map of a 7-by-20 grid of cells. Each cell can be empty, contain a star, or act as a wall. The goal is to navigate the bird across the map from left to right and collect as many stars as possible. However, colliding with a wall or reaching the (upper and lower) boundaries leads to the end of the game. An example map is displayed in Figure 1, which splits into three phases: the first phase contains almost only stars and no walls, the second phase contains almost only walls and very few stars, and the third phase contains both stars and walls.

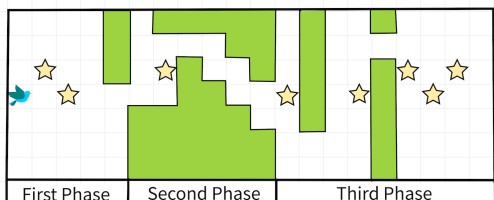

Figure 1: Flappy Bird environment: player needs to navigate the bird to avoid walls and collect stars.

We define the state space as the current locations of the bird on the grid, represented by coordinates $(x, y) \in \mathbb{Z}^2$, with a total of $7 \times 20 = 140$ states. Regarding the action space, we define it as $\mathcal{A} = \{\text{Up, Up-Up, Down}\}$. Each action causes the bird to move forward by one cell. In addition, the "Up" action moves the bird one cell upwards, the "Up-Up" action moves it two cells upwards, and the "Down" action moves it one cell downwards. The MDP has a reward as a function of state only. We will get a reward of 1 when the current state (location) has a star and otherwise 0. To model human behavior, we consider two sub-optimal human policies: **Policy Greedy**, which prioritizes collecting stars in the next column, and **Policy Safe**, which focuses on avoiding walls in the next column. If there is no preferred action available, both policies maintain a horizontal zig-zag line by alternating between "Up" and "Down". For adherence level $\theta$, we assume for all $s \in \mathcal{S}$ and $h = 1, ..., H$, the human will adhere to the advice with probability 0.9 except the aggressive advice "Up-up" (which moves too fast vertically) with adherence level 0.7. We compare the following algorithms:

- UCB-ADherence (UCB-AD): Algorithm 1 that finds the $\epsilon$-optimal advice policy.

- RFE-ADvice (RFE-AD): Algorithm 3, a variant of RFE-$\beta$ that finds the $\epsilon$-optimal policy.

- RFE-$\beta$: Algorithm 2 that outputs pertinent advice policy by exploring then planning.

- RFE-CMDP: A variant of RFE-$\beta$ that solves the CMDP (7) after exploring.

Figure 2a and 2b present the results for the two algorithms UCB-AD and RFE-AD for the environment $\mathcal{E}_1$. It also includes the state-of-the-art algorithm EULER (Zanette and Brunskill, 2019) that achieves a generic minimax optimal regret. From the regret plot, UCB-AD outperforms both RFE-AD and EULER. This advantage is attributed to UCB-AD's effective utilization of the information and structure of the underlying MDP. These results also show that our tailored algorithms UCB-AD and RFE-AD are much more efficient than directly applying problem-agnostic RL algorithms in the adherence model. We further test UCB-AD with different $\theta$'s: with $\theta_1$, $\theta(a, s) \equiv 0.8$ and with $\theta_2$, $\theta(a, s) \equiv 0.4$. Figure 2c shows the relationship between the regret of UCB-AD and $\theta$: for both policies, UCB-AD can achieve smaller regret with higher $\theta$. Intuitively, a high adherence level implies

a high probability of following the advice instead of taking $\pi^{\text{H}}$, which will reduce the regret caused by the suboptimality of $\pi^{\text{H}}$.

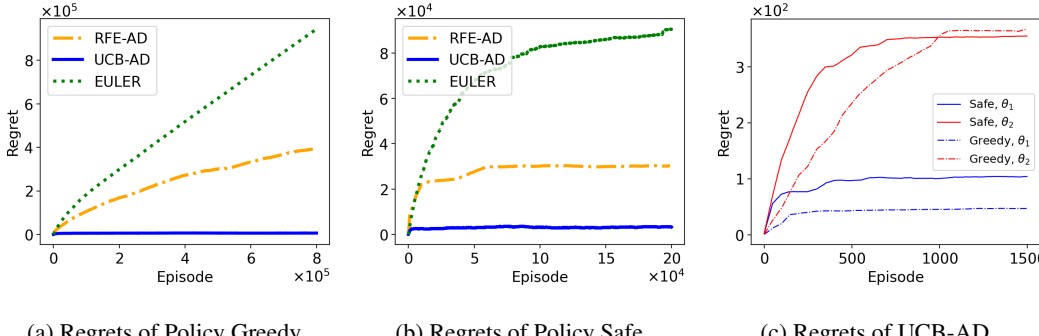

(a) Regrets of Policy Greedy.     (b) Regrets of Policy Safe.     (c) Regrets of UCB-AD.

Figure 2: The regrets for learning the optimal advice for Policy Greedy and Policy Safe. Figure 2a, 2b show the regrets of RFE-AD, UCB-AD, and EULER for two policies respectively. Figure 2c shows the regrets of UCB-AD for two policies under different $\theta$'s.

Figure 3 summarizes results for three policies under the environment $\mathcal{E}_2$, namely RFE-$\beta$, RFE-CMDP, and UC-CFH, a provably convergent CMDP algorithm (Kalagarla et al., 2021), under Policy Safe. In Figure 3a, we see that RFE-$\beta$ exhibits convergence for different $\beta$'s, and this empirically corroborates the theoretical finding. In Figure 3b, we compare RFE-CMDP and UC-CFH under a simpler environment with the advice budget being 1 ($D = 1$). We observe that RFE-CMDP shows a marginal performance advantage over UC-CFH in terms of the convergence rate. More importantly, Figure 3c shows by only using the estimated transition kernel after learning for $D = 1$ (Figure 3b), RFE-CMDP is able to obtain near-optimal policy for problem instances with different advice budgets ($D = 2, 3, 4$ and $5$). However, UC-CFH fails to explore the whole transition kernel sufficiently and can only output the near-optimal policy for the original problem instance. Moreover, RFE-CMDP is more sample efficient with respect to the advice budget, because for UC-CFH, we have to run multiple times with different advice budget parameters to get a near-optimal policy for all of them.

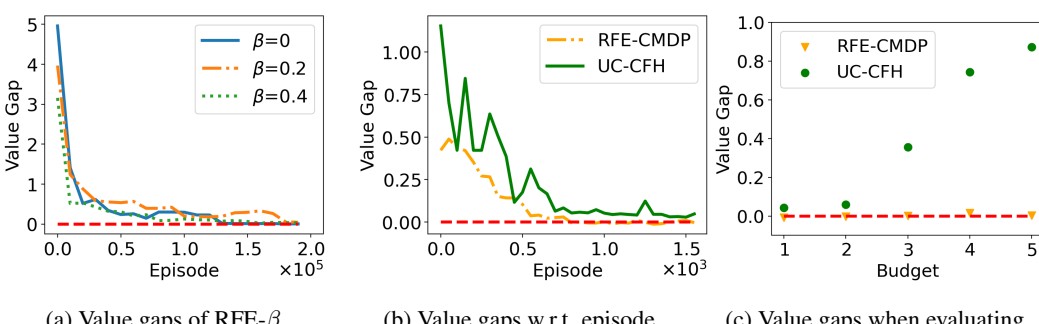

(a) Value gaps of RFE-$\beta$.     (b) Value gaps w.r.t. episode.     (c) Value gaps when evaluating.

Figure 3: The performances of making pertinent advice. The value gap is defined as the difference between the value of current policy and the optimal values, with the red dashed line as the benchmark for 0 loss of the policy. Figure 3a shows the convergence of RFE-$\beta$ under difference $\beta$'s. Figure 3b compares the convergences of RFE-CMDP and UC-CFH. Figure 3c evaluates performance of policy learned from learning episodes in Figure 3b.

Lastly, we show that RFE-$\beta$ is capable of generating pertinent advice for different policies. Figure 4 displays representative trajectories of two policies playing the game while receiving guidance from the machine, which follows $\hat{\pi}_\beta$ trained in the experiment of Figure 2. By setting $\beta = 0.3$, the machine outputs a policy that only gives advice when necessary: Since Policy Greedy behaves well in the first phase, the machine almost only gives advice in the second phase and the third phase; Similarly, the machine almost only gives advice in the first phase and the third phase, and choose to defer most of the time when Policy Safe is in the second phase.

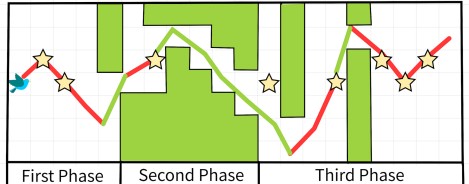

(a) Trajectory of Policy Greedy.

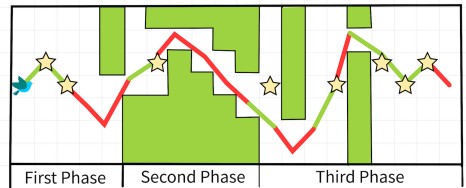

(b) Trajectory of Policy Safe.

Figure 4: Typical trajectories of two policies' types. The red color means the machine defers and the green color means the machine advises.

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
