# A  PROOFS

In this appendix, we present the proofs for Section 2 in A.1, the proofs and other details for Algorithm 1 in Section A.2, and the proofs and other details for Algorithm 2 in Section A.3.

## A.1  PROOFS FOR SECTION 2

*Proof of Proposition 2.* To show $V^*(s|\theta_1) \geq V^*(s|\theta_2)$, it suffices to prove that there exists $\pi \in \Pi$ such that $V^\pi(s|\theta_1) \geq V^*(s|\theta_2)$. Denote $\pi^{*,\theta_2}$ the optimal deterministic policy such that $\pi^{*,\theta_2} = \arg\max_{\pi \in \Pi} V^\pi(s|\theta_2)$. Based on $\pi^{*,\theta_2}$, we are going to construct a randomized policy $\tilde{\pi}^{\theta_1}$ such that $V^{\tilde{\pi}^{\theta_1}}(s|\theta_1) = V^*(s|\theta_2)$, and in the end, we will conclude the proof by showing $V^*(s|\theta_1) \geq V^{\tilde{\pi}^{\theta_1}}(s|\theta_1)$.

For any $h \in [H]$ and $s \in \mathcal{S}$, with a slight abuse of notation denote $a = \pi_h^{*,\theta_2}(s)$, the deterministic action of policy $\pi_h^{*,\theta_2}$. If $a = $ defer, the randomized advice policy $\tilde{\pi}^{\theta_1}$ is defined as $\tilde{\pi}_h^{\theta_1}(s) = $ defer; if $a \neq $ defer, we have

$$
\tilde{\pi}_h^{\theta_1}(s) = \begin{cases} a, & \text{with probability } \frac{\theta_2(s,a) - \pi^{\text{H}}(a|s)}{\theta_1(s,a) - \pi^{\text{H}}(a|s)} \\ \text{defer}, & \text{with probability } 1 - \frac{\theta_2(s,a) - \pi^{\text{H}}(a|s)}{\theta_1(s,a) - \pi^{\text{H}}(a|s)} \end{cases}
$$

where we have assumed that $\theta_i(s, a) \geq \pi_h^{\text{H}}(a|s)$, i.e, the advice has no negative effect on the human's probability of taking action $a$ at state $s$ for $i = 1, 2$. Next, we have to verify $V^{\tilde{\pi}^{\theta_1}}(s|\theta_1) = V^*(s|\theta_2)$ by showing that for any $(a, s, h) \in (\bar{\mathcal{A}}, \mathcal{S}, [H])$,

$$
P_h^{\tilde{\pi}^{\theta_1}}(a^{\text{H}} = a|s, \theta_1) = P_h^{\pi_h^{*,\theta_2}}(a^{\text{H}} = a|s, \theta_2). \tag{9}
$$

To see this, for $a = \pi_h^{*,\theta_2}(s)$ and $a \neq $ defer, we have

$$
\begin{aligned}
P_h^{\tilde{\pi}^{\theta_1}}(a^{\text{H}} = a|s, \theta_1) &= \frac{\theta_2(s,a) - \pi_h^{\text{H}}(a|s)}{\theta_1(s,a) - \pi_h^{\text{H}}(a|s)} \theta_1(s,a) + \left(1 - \frac{\theta_2(s,a) - \pi_h^{\text{H}}(a|s)}{\theta_1(s,a) - \pi_h^{\text{H}}(a|s)}\right) \pi_h^{\text{H}}(a|s) \\
&= \theta_2(s,a) \\
&= P_h^{\pi_h^{*,\theta_2}}(a^{\text{H}} = a|s, \theta_2).
\end{aligned}
$$

For $a' \neq \pi_h^{*,\theta_2}(s)$ and $a \neq $ defer, we have

$$
\begin{aligned}
P_h^{\tilde{\pi}^{\theta_1}}(a^{\text{H}} = a'|s, \theta_1) &= \frac{\theta_2(s,a) - \pi_h^{\text{H}}(a|s)}{\theta_1(s,a) - \pi_h^{\text{H}}(a|s)} \cdot (1 - \theta_1(s,a)) \cdot \frac{\pi_h^{\text{H}}(a'|s)}{1 - \pi_h^{\text{H}}(a|s)} \\
&\quad + \left(1 - \frac{\theta_2(s,a) - \pi_h^{\text{H}}(a|s)}{\theta_1(s,a) - \pi_h^{\text{H}}(a|s)}\right) \pi_h^{\text{H}}(a'|s) \\
&= (1 - \theta_2(s,a)) \frac{\pi_h^{\text{H}}(a'|s)}{1 - \pi_h^{\text{H}}(a|s)} \\
&= P_h^{\pi_h^{*,\theta_2}}(a^{\text{H}} = a'|s, \theta_2).
\end{aligned}
$$

For the last case, it is obvious that if $\pi_h^{*,\theta_2}(s) = $ defer, the dynamics of choosing defer are independent of $\theta$, and this concludes proving (9).

By showing (9), we know that $V^{\tilde{\pi}^{\theta_1}}(s|\theta_1) = V^{\pi_h^{*,\theta_2}}(s|\theta_2) = V^*(s|\theta_2)$. Because we are working on a finite-horizon discrete MDP, from Bellman's equation, we know that the optimal value functions for the class of deterministic policies will be the same as those for the class of random policies. Therefore, we have $V^*(s|\theta_1) \geq V^{\tilde{\pi}^{\theta_1}}(s|\theta_1)$, and this concludes the proof. □

*Proof of Proposition 1.* From the Bellman's equation, we know that if $a = \arg\max_{a \in \bar{\mathcal{A}}} Q_{h,\beta}^*(s, a)$ and $a \neq $ defer, we have $r_h^{\text{M}}(s, a) - \beta + \sum_{s' \in \mathcal{S}} p_h^{\text{M}}(s'|s, a) V_{h+1}^*(s') \geq V_{h,\beta}^{\pi^{\text{H}}}(s)$, and therefore

$$
\beta \leq r_h^{\text{M}}(s, a) + \sum_{s' \in \mathcal{S}} p_h^{\text{M}}(s'|s, a) V_{h+1,\beta}^*(s') - V_{h,\beta}^{\pi^{\text{H}}}(s).
$$

By observing that $V_\beta^{\pi^H} = V^{\pi^H}$ (always deferring has no penalty), and $V_\beta^* \le V^*$, we have

$$\beta \le r_h^M(s,a) + \sum_{s' \in \mathcal{S}} p_h^M(s'|s,a) V_{h+1}^*(s') - V_h^{\pi^H}(s)$$

$$= Q_h^*(s,a) - V_h^{\pi^H}(s).$$

$\square$

## A.2 Supplementary Materials for Algorithm 1

In this section, we first formally state the parameter updating rules in Algorithm 1 and define the related functions and parameters within the algorithm in Section A.2.1. Subsequently, in Section A.2.2, we present several useful lemmas to sketch the proof for Theorem 1. Then, in Section A.2.3, we prove all statements under $\mathcal{E}_1$.

### A.2.1 Notations of Algorithm 1

In this section, we first state the parameter updating rules in Algorithm 1 under $\mathcal{E}_1$ as follows:

$$\hat{\theta}^t(s,a) = \frac{1}{n^t(s,a)} \sum_{i=1}^t \sum_{h=1}^H \mathbb{I}(s_h^i = s, a_h^{M,i} = a, a_h^{H,i} = a),$$

where $n^t(s,a) = \sum_{i=1}^t \sum_{h=1}^H \mathbb{I}(s_h^i = s, a_h^{M,i} = a)$ for all $s \in \mathcal{S}$, $a \in \mathcal{A}$ and $t \le T$.

$$\bar{\theta}^t(s,a) = \min\left\{ 1, \hat{\theta}^t(s,a) + \frac{C\left(\hat{\theta}^t(s,a), n^t(s,a), T, \delta\right)}{\sqrt{n^t(s,a)}} \right\},$$

where

$$C(\theta, n, T, \delta) = \min\left\{ 2\sqrt{\log(12SAT/\delta)}, \sqrt{2\theta(1-\theta)\log\frac{12SAT}{\delta}} + \frac{7\sqrt{n}}{3n-1} \cdot \log\frac{12SAT}{\delta}, \right.$$

$$\left. \left( \frac{1 + \sqrt{1 + 4\left(\max\left\{0, \sqrt{\theta(1-\theta)} - \sqrt{\frac{2\log(SAT/\delta)}{n-1}}\right\}\right)^2}}{2} - \theta \right) \cdot \sqrt{n} \right\}.$$

Here, by definition $\bar{\theta}^t(s,a)$ becomes the largest element in the following set

$$\{\theta \in [0,1] : |\theta - \hat{\theta}^t(s,a)| \le 2\sqrt{\frac{\log(12SAT/\delta)}{n^t(s,a)}}, \tag{10}$$

$$|\theta - \hat{\theta}^t(s,a)| \le \sqrt{\frac{2\hat{\theta}^t(s,a)(1 - \hat{\theta}^t(s,a))}{n^t(s,a)} \log\frac{12SAT}{\delta}} + \frac{7}{3n^t(s,a) - 1} \cdot \log\frac{12SAT}{\delta}$$

$$|\sqrt{\theta(1-\theta)} - \sqrt{\hat{\theta}^t(s,a)^t(1 - \hat{\theta}^t(s,a))}| \le \frac{2\log(SAT/\delta)}{n^t(s,a) - 1} \},$$

for any state-action pair $(s,a) \in \mathcal{S} \times \mathcal{A}$ and $t \le T$. Lemma 1, which we will introduce later, states that the true adherence level $\theta^t(s,a)$ is also in this set with high probability, and thus, $\bar{\theta}^t(s,a)$ is an upper bound for the true adherence level for all $s, a, h$ and $t \le T$.

We point out that our algorithm needs to have a pre-determined episode upper bound $T$ as the input. The reason is that Algorithm 1 updates the estimation of the rewards and the adherence levels at the end of all episodes, which requires us to take a union probability bound from Hoeffding's inequality for $T$ episodes. In order to alleviate the effect of $T$ in the probability union bound, we need to consider $T$ in the design of the algorithm.

In this section, we prove Theorem 1 for Algorithm 1 under $\mathcal{E}_1$, where the human policy is known while the adherence level is unknown. While the proof is inspired by the proof of Theorem 1 in Dann and Brunskill (2015), our proof can achieve a better sample complexity bound than theirs in our setting, where the transition probability $p_h^{\mathtt{M}}$ is known but depends on the time horizon for all $h \in [H]$, and the adherence level $\theta$ is unknown but independent of $h$. Specifically, applying Theorem 1 in Dann and Brunskill (2015) can only achieve an $O(H^3 S^2 A/\epsilon^2)$ sample complexity bound. On the contrary, we establish an $O(H^2 S^2 A/\epsilon^2)$ sample complexity bound, which improves the order of the length of the time horizon in the bound.

First, we seek to build an analysis framework for a simpler problem: the reward function is deterministic. This is because if the reward function is stochastic, we can adopt the idea used in the proof of Lemma 3 and analyze an additional error term for the estimation of the reward. By doing so, we can obtain a similar result with the same order with respect to $H$, $S$, and $A$ but with different universal constants.

Therefore, in the following, we only analyze the case where the reward function of the machine and the human policy is deterministic. Then, the challenge lies in estimating the adherence levels. Recall that the estimator for the adherence level is

$$\hat{\theta}^t(s,a) = \frac{1}{n^t(s,a)} \sum_{i=1}^{t} \sum_{h=1}^{H} \mathbb{I}(s_h^i = s, a_h^{\mathtt{M},i} = a, a_h^{\mathtt{H},i} = a),$$

where $n^t(s,a) = \sum_{i=1}^{t} \sum_{h=1}^{H} \mathbb{I}(s_h^i = s, a_h^{\mathtt{M},i} = a)$ for all $s \in \mathcal{S}$, $a \in \mathcal{A}$ and $t \leq T$.

Next, we briefly summarize our proof. In general, we follow a similar proof structure as in Theorem 1 of Dann and Brunskill (2015). However, our setting features two differences from theirs as listed below.

**Higher updating frequency.**

The first difference is that we update the estimation of the transition kernels, or the adherence levels at the end of each episode for all states and actions, while Dann and Brunskill (2015) only update the estimation once after several episodes for only one state-action pair. Although a slower updating rule makes their algorithm slow, the problem of frequent updates is that we are exposed to a larger probability of observing some outliers such that the true adherence level might not be captured by the confidence set (10). Therefore, the probability bound developed by Lemma 1 in Dann and Brunskill (2015) will fail in our setting.

**Non-Stationarity.**

Our setting is non-stationary in the sense that the transition probability $p_h^{\mathtt{M}}$ is different for different $h = 1, ..., H$ even if the adherence level is independent of the horizon. However, Dann and Brunskill (2015) consider a stationary setting where the transition kernels depend only on the state-action pairs and are the same for all different $h = 1, ..., H$. As a result, we cannot directly apply their Lemma C.5 to bound several parameters (those two parameters are $c_1(s,a)$ and $c_2(s,a)$ on page 20 in Dann and Brunskill (2015)) directly.

To address those two problems, we show the following two lemmas. For the first problem, Lemma 1 states that even with higher frequent updates, the confidence set (10) can still cover the true adherence level with high probability, and thus, Proposition 2 implies that our estimated $Q$-values are upper bounds for the corresponding true $Q$-values for all empirical optimal policy $\hat{\pi}^t$ at all episodes $t \leq T$ for any pre-determined constant $T$ in Algorithm 1.

**Lemma 1.** *For any fixed $T \geq 0$, with probability at least $1 - \frac{\delta}{2}$, the set (10) contains the true adherence level $\theta(s,a)$ for any state-action pairs $(s,a) \in \mathcal{S} \times \mathcal{A}$ and episode $t \leq T$.*

For the second problem, Lemma 2 transfers the concentration of the non-stationary transition kernels to that of the stationary adherence levels that are independent of the horizon $H$ and, establishes a Bernstein-like bound for all non-stationary transition kernels.

**Lemma 2.** *The following inequalities*

$$\left| p_h^{\mathtt{M}}(s'|s,a) - \bar{p}_h^{\mathtt{M},t}(s'|s,a) \right| \leq \left| \left( \pi_h(s'|s,a) - \frac{\sum\limits_{a' \neq a} \pi_h(s'|s,a')\pi_h^{\mathtt{H}}(a'|s)}{1 - \pi_h^{\mathtt{H}}(a|s)} \right) (\theta(s,a) - \hat{\theta}^t(s,a)) \right|$$

$$\leq \sqrt{\frac{8\hat{p}_h^{\mathtt{M},t}(s'|s,a)(1 - \hat{p}_h^{\mathtt{M},t}(s'|s,a))}{n^t(s,a)} \log(12SAT/\delta)} + \frac{26}{3n^t(s,a) - 3}\log(12SAT/\delta)$$

*hold with probability no less than $1 - \delta$ for all $s, s' \in \mathcal{S}$, $a \in \mathcal{A}$ and $t \leq T$.*

Then, based on those two above lemmas, the proof can be shown in a similar approach as in Dann and Brunskill (2015).

Before we prove the main theorem, we need some notations. As in Dann and Brunskill (2015), denote $w_t(s,a)$ the expected visitation frequency of the $(s,a)$-pair under policy $\pi^t$, i.e.,

$$w_t(s,a) = \sum_{h=1}^{H} P\left(s_h = s, \pi_h^t(s_h) = a\right).$$

Next, we denote $\iota_t(s,a)$ the importance of $(s,a)$: its relative weight compared to $w_{\min} := \frac{\epsilon}{4H|\mathcal{S}|}$ on a log-scale

$$\iota_t(s,a) := \min \left\{ z_i : z_i \geq \frac{w_t(s,a)}{w_{\min}} \right\} \in \{0, 1, 2, 4, 8, \ldots\} \quad \text{where } z_1 = 0 \text{ and } z_i = 2^{i-2} \quad \forall i = 2, 3, \ldots$$

Intuitively, $\iota_t(s,a)$ is an integer indicating the influence of the state-action pair on the value function of $\pi^t$. Similarly, we define the knownness

$$\kappa_t(s,a) := \max \left\{ z_i : z_i \leq \frac{n_t(s,a)}{m w_t(s,a)} \right\} \in \{0, 1, 2, 4, \ldots\},$$

which indicates how often $(s,a)$ has been observed relative to its importance. The constant $m$ is defined as

$$m = 512(\log_2 \log_2 H)^2 \frac{CH2}{\epsilon^2} \log_2^2 \left( \frac{8H^2 S^2}{\epsilon} \right) \log \left( \frac{6CSA \log_2^2(4S^2 H2/\epsilon)}{\delta} \right),$$

where $C = \max_{s \in \mathcal{S}, a \in \bar{\mathcal{A}}} C(s,a)$, and $C(s,a)$ denotes possible successor states of state $s$ and action $a$ for $s \in \mathcal{S}$ and $a \in \mathcal{A}$. Thus, we also have $C \leq S$. We can now categorize $(s,a)$-pairs into subsets

$$X_{t,\kappa,\iota} := \{(s,a) \in X_t : \kappa_t(s,a) = \kappa, \iota_t(s,a) = \iota\} \quad \text{and} \quad \bar{X}_t = \mathcal{S} \times \mathcal{A} \backslash X_t$$

where $X_t = \{(s,a) \in \mathcal{S} \times \mathcal{A} : \iota_t(s,a) > 0\}$ is the active set and $\bar{X}_t$ the set of state-action pairs that are very unlikely under the current policy.

The proof can be summarized in a few steps:

1. The true MDP is in the confidence set of MDPs (those with adherence level in (10)) for all episodes $t < T$ with probability at least $1 - \delta/2$ (we can ensure this property by Lemma 1).

2. In every episode $t$, the optimistic $Q$-functions $\hat{V}^{\hat{\pi}^t}(\cdot|\bar{\theta}^t)$ is higher than $V^*(\cdot|\theta)$ at least $1 - \delta/2$, which is ensured by Proposition 2.

3. If $m = \tilde{\Omega}\left(\frac{H^2}{\epsilon} \ln \frac{|\mathcal{S}|}{\delta}\right)$ (which is true by our definition of $m$), the number of episodes with $|X_{t,\kappa,\iota}| > \kappa$ for some $\kappa$ and $\iota$ are bounded by $\tilde{O}(|\mathcal{S} \times \mathcal{A}|m)$ with probability at least $1 - \delta/2$. To show this step, we can apply Lemma 2 in Dann and Brunskill (2015).

4. If $|X_{t,\kappa,\iota}| \leq \kappa$ for all $\kappa$, $\iota$, i.e., relevant state-action pairs are sufficiently known and $m = \tilde{\Omega}\left(\frac{CH^2}{\epsilon^2} \ln \frac{1}{\delta_1}\right)$, then the optimistic values $\hat{V}^{\hat{\pi}^t}(s_1|\bar{\theta}^t)$ and $V^{\hat{\pi}^t}(s_1|\theta^t)$ are $\epsilon$-close to the true MDP value. Together with part 2, we get that with high probability, the policy $\hat{\pi}^t$ is $\epsilon$-optimal in this case. To prove this, we can use our Lemma 2 combined with Lemma 3 in Dann and Brunskill (2015)

5. From parts 3 and 4 , we can show that with probability $1 - \delta$, at most $\tilde{O}\left(\frac{C|\mathcal{S}\times\mathcal{A}|H^2}{\epsilon^2}\ln\frac{1}{\delta}\right) = \tilde{O}\left(\frac{H^2S^2A}{\epsilon^2}\ln\frac{1}{\delta}\right)$ episodes are not $\epsilon$-optimal.

In the following, we show the proof of Theorem 1, Lemmas 1 and 2.

### A.2.3 PROOF RELATED TO THEOREM 1

In this section, we first prove Theorem 1, and then prove Lemmas 1 and 2 that are used in the proof.

*Proof of Theorem 1.* In our case, the statements of Lemmas 2 and 3 in Dann and Brunskill (2015) still hold. Lemma 2 can be shown without any modifications, and for their Lemma 3, we can show the same statement by applying Lemma 2 to establish $c_1(s, a)$ and $c_2(s, a)$ on page 20 of Dann and Brunskill (2015) for our non-stationary case.

Then, by Lemma 2 in Dann and Brunskill (2015), we can bound the number of episodes satisfying $|X_{t,\kappa,\iota}| > \kappa$ for some $\kappa, \iota$ by $6mSA \cdot \log_2\frac{4H^2S}{\epsilon}\log_2 S = O(C \cdot H^2SA/\epsilon^2)$ with probability at least $1 - \delta/2$. Their Lemma 3 states that

$$\left|V^{\hat{\pi}_t}(s_1) - \hat{V}^{\hat{\pi}_t}(s_1)\right| < \epsilon \tag{11}$$

for all other episodes. In addition, combining Lemma 1 and Proposition 2, we know that

$$\hat{V}^{\hat{\pi}^t} \geq V^{\pi^*} \geq V^{\hat{\pi}^t} \tag{12}$$

holds with probability at least $1 - \delta/2$. Thus, we can draw the conclusion that $1 - \delta$, with at least $T - O(C \cdot H^2SA/\epsilon^2)$ episodes, the corresponding policy $\hat{\pi}^t$ satisfies

$$V^{\hat{\pi}^t}(s_1) + \epsilon \geq \hat{V}^{\hat{\pi}^t}(s_1) \geq V^{\pi^*}(s_1) \geq V^{\hat{\pi}^t}(s_1),$$

which implies that the corresponding $\hat{\pi}^t$ is $\epsilon$-optimal. Here, the first inequality comes from (11), and others come from (12). Moreover, recall that $C$ is the maximum number of possible successor states from any state and action pair, implying $C \leq S$. Thus, at most $O(C \cdot H^2SA/\epsilon^2) = O(H^2S^2A/\epsilon^2)$ episodes are not $\epsilon$-optimal, as our statement of Theorem 1. $\square$

*Proof of Lemma 1.* The proof is similar to the proof of Lemma 1 in Dann and Brunskill (2015). Given the total visiting number $n^t(s, a)$ of a state-action pair $(s, a)$ and the corresponding visiting horizons and episodes $\{(h_l, t_l)\}_{l=1}^{n^t(s,a)}$ such that $s_h^t = s$ and $a_h^{\text{M},i} = a$, we have

$$\mathbb{E}\left[\frac{1}{n^t(s,a)}\sum_{l=1}^{n^t(s,a)}\mathbb{I}(s_{h_l}^{t_l} = s, a_{h_l}^{\text{M},t_l} = a, a_h^{\text{H},t_l} = a)\right] = \theta(s, a) \tag{13}$$

by the definition of $\theta(s, a)$ (recall the definition $n^t(s, a) = \sum_{i=1}^{t}\sum_{h=1}^{H}\mathbb{I}(s_h^i = s, a_h^{\text{M},i} = a)$). Then, by the Azuma–Hoeffding's inequality, with given $n^t(s, a)$ and $\{(h_l, t_l)\}_{l=1}^{n^t(s,a)}$, the following inequality holds with probability no less than $1 - \frac{\delta}{12SAT}$

$$\left|\hat{\theta}^t(s, a) - \theta(s, a)\right| = \left|\frac{1}{n^t(s,a)}\sum_{i=1}^{t}\sum_{h=1}^{H}\mathbb{I}(s_h^i = s, a_h^{\text{M},i} = a, a_h^{\text{H},i} = a) - \theta(s, a)\right| \leq 2\sqrt{\frac{\log(12SAT/\delta)}{n^t(s,a)}} \tag{14}$$

holds for all $s \in \mathcal{S}$, $a \in \mathcal{A}$ and $t$. Here, the first step comes from the definition of $\hat{\theta}^t(s, a)$, and the second line comes from Hoeffding's inequality and (13). Consequently, taking a union bound for all $s, a$ and $t \leq T$, we have that with probability no less than $1 - \frac{\delta}{12}$

$$\left|\hat{\theta}^t(s, a) - \theta(s, a)\right| \leq 2\sqrt{\frac{\log(12SAT/\delta)}{n^t(s,a)}}$$

holds for all $(s,a) \in \mathcal{S} \times \mathcal{A}$ at the end of any episode $t \leq T$. Similarly, by applying Bernstein's inequality and taking union bound, we have

$$\left|\hat{\theta}^t(s,a) - \theta(s,a)\right| \leq \sqrt{\frac{2\theta(s,a)(1-\theta(s,a))\log(12SAT/\delta)}{\sqrt{n^t(s,a)}}} + \frac{1}{3n^t(s,a)}\log(6SAT/\delta), \quad (15)$$

which holds also for all $s, a$ and $t \leq T$ with probability no less than $1 - \frac{\delta}{12}$. In addition, by applying Theorem 10 in Dann and Brunskill (2015), we can have that with probability no less than $1 - \frac{\delta}{12HST}$,

$$\left|\sqrt{\theta(s,a)(1-\theta(s,a))} - \sqrt{\hat{\theta}^t(s,a)(1-\hat{\theta}^t(s,a))}\right| \leq \sqrt{\frac{2\log(12SAT)}{n^t(s,a)}}. \quad (16)$$

Thus, combining the three probability bounds (14), (15), and (16), we finally have that with probability no less than $1 - \frac{\delta}{4}$, the true adherence level $\theta(s,a)$ satisfies all three inequalities in (10) for all $s, a$ and $t \leq T$.

Additionally, we remark that our construction in Algorithm 1 is compatible with the setting where the reward function $r^{\text{M}}$ is unknown and can be random. Even if the reward function is random, we can follow the same proof as for the adherence level to show that the estimated average reward functions in Algorithm 1 is an upper bound of the true average reward function with probability no less than $1 - \frac{\delta}{4}$, and the difference between them is in the order of $O(1/\sqrt{n^t(s,a)})$ for all $s, a$ and $t \leq T$. Therefore, with probability no less than $1 - \frac{\delta}{2}$, our estimated reward function and the adherence level are larger than the corresponding true values, and the convergence rate is $O(1/\sqrt{n^t(s,a)})$ for all $s, a$ and $t \leq T$. $\square$

*Proof of Lemma 2.* By applying the definition of $p_h^{\text{M}}$ and $p_h^{\text{M},t}$, we have

$$p_h^{\text{M}}(s'|s,a) = \left(\pi_h(s'|s,a) - \frac{\sum\limits_{a' \neq a}\pi_h(s'|s,a')\pi_h^{\text{H}}(a'|s)}{1 - \pi_h^{\text{H}}(a|s)}\right)\theta(s,a) + \frac{\sum\limits_{a' \neq a}\pi_h(s'|s,a')\pi_h^{\text{H}}(a'|s)}{1 - \pi_h^{\text{H}}(a|s)},$$

$$\hat{p}_h^{\text{M},t}(s'|s,a) = \left(\pi_h(s'|s,a) - \frac{\sum\limits_{a' \neq a}\pi_h(s'|s,a')\pi_h^{\text{H}}(a'|s)}{1 - \pi_h^{\text{H}}(a|s)}\right)\bar{\theta}^t(s,a) + \frac{\sum\limits_{a' \neq a}\pi_h(s'|s,a')\pi_h^{\text{H}}(a'|s)}{1 - \pi_h^{\text{H}}(a|s)}.$$

$$(17)$$

Then, inequalities in (17) imply the first inequality in Lemma 2 for all $s', s \in \mathcal{S}$, $a \in \mathcal{A}$ and $t$.

Next, we prove the second inequality. For reading convenience, we let $\zeta_1$ and $\zeta_2$ be

$$\zeta_1 = \left(\pi_h(s'|s,a) - \frac{\sum\limits_{a' \neq a}\pi_h(s'|s,a')\pi_h^{\text{H}}(a'|s)}{1 - \pi_h^{\text{H}}(a|s)}\right), \quad \zeta_2 = \frac{\sum\limits_{a' \neq a}\pi_h(s'|s,a')\pi_h^{\text{H}}(a'|s)}{1 - \pi_h^{\text{H}}(a|s)},$$

for any fixed state-action pair $s, a$ and fixed $t$. Here, by definition, we have $\zeta_1 \in [-1, 1]$ and $\zeta_2 \in [0, 1]$. Then, we have

$$\left|p_h^{\text{M}}(s'|s,a) - \hat{p}_h^{\text{M},t}(s'|s,a)\right| = |\zeta_1||\theta(s,a) - \bar{\theta}^t(s,a)|.$$

Now, we first show

$$|\zeta_1|\bar{\theta}^t(s,a)(1 - \bar{\theta}^t(s,a)) \leq \hat{p}_h^{\text{M},t}(s'|s,a)(1 - \hat{p}_h^{\text{M},t}(s'|s,a)). \quad (18)$$

If $\zeta_1 \geq 0$, (18) is equivalent to

$$(\zeta_1^2 - \zeta_1)(\bar{\theta}^t(s,a))^2 + 2\zeta_1\zeta_2\bar{\theta}^t(s,a) + \zeta_2^2 - \zeta_1 \leq 0,$$

which can be obtained by checking the non-positivity of the discriminant for the quadratic equation. Specifically, the discriminant is

$$-4\zeta_1 - 4\zeta_2 + 4\zeta_1\zeta_2^2 + 4\zeta_1^2\zeta_2,$$

which is no more than 0 since $\zeta_1, \zeta_2 \leq 1$. If $\zeta_1 \leq 0$, we can similarly prove that the discriminant is still non-positive. Therefore, (18) holds. Then, by Lemma C.5 in Dann and Brunskill (2015), we have on the event of Lemma 1,

$$|\theta(s,a) - \bar{\theta}^t(s,a)| \leq \sqrt{\frac{8\bar{\theta}^t(s,a)(1-\bar{\theta}^t(s,a))}{n^t(s,a)} \log(12SAT/\delta)} + \frac{26}{3n^t(s,a) - 3} \log(12SAT/\delta). \tag{19}$$

Combining (19) and (18), and plugging them into (17), we arrive to

$$\begin{aligned}
\left|p_h^{\mathtt{M}}(s'|s,a) - \hat{p}_h^{\mathtt{M},t}(s'|s,a)\right| &= |\zeta_1||\theta(s,a) - \bar{\theta}^t(s,a)| \\
&\leq |\zeta_1|\sqrt{\frac{8\bar{\theta}^t(s,a)(1-\bar{\theta}^t(s,a))}{n^t(s,a)} \log(12SAT/\delta)} + \frac{26|\zeta_1|}{3n^t(s,a)-3} \log(12SAT/\delta) \\
&\leq \sqrt{\frac{8|\zeta_1|\bar{\theta}^t(s,a)(1-\bar{\theta}^t(s,a))}{n^t(s,a)} \log(12SAT/\delta)} + \frac{26}{3n^t(s,a)-3} \log(12SAT/\delta) \\
&\leq \sqrt{\frac{8\hat{p}_h^{\mathtt{M},t}(s'|s,a)(1-\hat{p}_h^{\mathtt{M},t}(s'|s,a))}{n^t(s,a)} \log(12SAT/\delta)} + \frac{26}{3n^t(s,a)-3} \log(12SAT/\delta),
\end{aligned}$$

where the first line comes from (17), the second line comes from (18), the third line comes from the fact that $|\zeta_1| \leq 1$, and the last line comes from (19). The proof of this inequality holds for all $s \in \mathcal{S}$, $a \in \mathcal{A}$ and $t \leq T$ on the event of Lemma 1, which holds with probability no less than $1 - \delta$. Thus, we finish the proof. $\square$

### A.3 Supplementary Materials for Algorithm 2

In this section, we first provide definitions of the estimations of the transition kernels and rewards in Algorithm 2, which can be found in Section A.3.1. In Section A.3.2, we will present the proof for Theorem 2 and Corollary 1. In section A.3.3, we prove important ancillary lemmas for Theorem 2.

#### A.3.1 Definitions in Algorithm 2

We first formally define the empirical estimation $\hat{p}_h^{\mathtt{M},t}$ in Algorithm 2 as follows:

$$\hat{p}_h^{\mathtt{M},t}(s'|s,a) = \frac{n_h^t(s,a,s')}{n_h^t(s,a)} \text{ if } n_h^t(s,a) > 0 \quad \text{and} \quad \hat{p}_h^{\mathtt{M},t}(s'|s,a) = \frac{1}{S} \quad \text{otherwise,}$$

where $n_h^t(s,a) = \sum_{i=1}^t \mathbb{I}\left\{\left(s_h^i, a_h^{\mathtt{M},i}\right) = (s,a)\right\}$ is the number of times in the first $t$ episodes at the time step $h$, state $s$, and the machine gives advice $a$; $n_h^t(s,a,s') = \sum_{i=1}^t \mathbb{I}\left\{\left(s_h^i, a_h^{\mathtt{M},i}, s_{h+1}^i\right) = (s,a,s')\right\}$ is the number of times at time $h$, state $s$, the machine gives advice $a$, and reached state $s'$ at time $h+1$ in the first $t$ episode. Similarly, the empirical reward is defined as

$$\hat{r}_h^{\mathtt{M},t}(s,a) = \frac{\sum_{i=1}^t r^i(s,a)\mathbb{I}\left\{\left(s_h^i, a_h^{\mathtt{M},i}\right) = (s,a)\right\}}{n_h^t(s,a)} \text{ if } n_h^t(s,a) > 0 \quad \text{and} \quad \hat{r}_h^{\mathtt{M},t}(s,a) = 0 \quad \text{otherwise.}$$

#### A.3.2 Proof of Theorem 2 and Corollary 1

To prove Theorem 2, we first state the algorithm and develop the corresponding sample complexity bound (with probability $1 - \delta$) for the no-penalty case, where the reward falls within the interval $[0, 1]$. Next, we show Theorem 2 by extending the above results into the case with the penalty $\beta \in (0, H)$, which can be addressed by scaling the penalized reward in the range $(0, H)$ to the range $[0, 1]$.

To solve the no-penalty case, as mentioned in the remark after Theorem 2, we change THRESHOLD from $\epsilon/H$ to $\epsilon/2$. The algorithm is summarized in Algorithm 3, and we define $\tau_1$ as the corresponding stopping time. In the following, we characterize the upper bound of $\tau_1$ and $V_1^*(s_1) - V_1^{\hat{\pi}^{\tau_1}}(s_1)$ when there is no penalty.

**Algorithm 3** : RFE-ADvice

1: Input: $\epsilon, \delta$
2: **Stage 1: Reward-free exploration**
3: Initialize $t = 1$, THRESHOLD = $\epsilon/2$, and $W_h^t(s, a) = H$ for all $(s, a) \in \mathcal{S} \times \mathcal{A}$
4: Compute $\pi^t$ so that $\pi_h^t(s) = \arg\max_{a \in \mathcal{A}} W_h^t(s, a)$ (see (5))
5: **while** $W_1^t(s_1, \pi^t(s_1)) + 4e\sqrt{W_1^t(s_1, \pi^t(s_1))} >$ THRESHOLD **do**
6:      Sample trajectory $z^t = \{s_1^t, a_1^{\text{M},t}, a_1^{\text{H},t}, r_1^t, \cdots, s_H^t, a_H^{\text{M},t}, a_H^{\text{H},t}, r_H^t\}$ following $\pi^t$
7:      update $t \leftarrow t + 1$, $\mathcal{D} \leftarrow \mathcal{D} \cup \{z^t\}$, $\hat{p}_h^{\text{M},t}(s'|s, a)$, $\hat{r}_h^{\text{M},t}(s, a)$, and $W_h^t(s, a)$
8: **end while**
9: **Stage 2: Policy identification**
10: Use planning algorithms to output optimal advice policy $\hat{\pi}^{\tau_1}$ for $(\mathcal{S}, \bar{\mathcal{A}}, H, \hat{p}^{\text{M}}, \hat{r}^{\text{M}})$

**Upper bounds of $\tau_1$ and $V_1^*(s_1) - V_1^{\hat{\pi}^{\tau_1}}(s_1)$.**

To establish the upper bound for $\tau_1$, our approach is similar to the proof of Theorem 1 in Ménard et al. (2021). First notice that

$$V_1^\star(s_1) - V_1^{\hat{\pi}^{\tau_1}}(s_1) = V_1^\star(s_1) - \widehat{V}_1^{\tau_1, \pi^\star}(s_1) + \widehat{V}_1^{\tau_1, \pi^\star}(s_1) - \widehat{V}_1^{\tau_1, \hat{\pi}^{\tau_1}}(s_1) + \widehat{V}_1^{\tau_1, \hat{\pi}^{\tau_1}}(s_1) - V_1^{\hat{\pi}^{\tau_1}}(s_1)$$
$$\leq \left| V_1^\star(s_1) - \widehat{V}_1^{\tau_1, \pi^\star}(s_1) \right| + \left| \widehat{V}_1^{\tau_1, \hat{\pi}^{\tau_1}}(s_1) - V_1^{\hat{\pi}^{\tau_1}}(s_1) \right|,$$

where the second inequality is because $\widehat{V}_1^{\tau_1, \pi^\star}(s_1) - \widehat{V}_1^{\tau_1, \hat{\pi}^{\tau_1}}(s_1) \leq 0$. Therefore, we need to show that the empirical MDP is close to the original MDP so that the value function of the same policy is bounded by $\epsilon/2$. To motivate this, recall the Bellman equation of the true $Q$-values and the empirical:

$$Q_h^\pi(s, a) = r_h^{\text{M}}(s, a) + \sum_{s' \in \mathcal{S}} p_h^{\text{M}}(s'|s, a) Q_{h+1}^\pi(s', \pi_{h+1}(s')), \tag{20}$$
$$\hat{Q}_h^{t,\pi}(s, a) = \hat{r}_h^{\text{M},t}(s, a) + \sum_{s' \in \mathcal{S}} \hat{p}_h^{\text{M},t}(s'|s, a) \hat{Q}_{h+1}^{t,\pi}(s', \pi_{h+1}(s')),$$

where $Q_{H+1}^\pi(s, a) = \hat{Q}_{H+1}^{t,\pi}(s, a) = 0$ for all staet $s$, action $a$, episode $t$, and policy $\pi$.

Denote $\hat{e}_h^{t,\pi}(s, a; r) = |\hat{Q}_h^{t,\pi}(s, a; \hat{r}) - Q_h^\pi(s, a; r)|$ the difference between the empirical and the real $Q$-value for the machine with respect to **any** policy $\pi$, state $s$, action $a$, reward $r$ ($\hat{r}$ is the sample estimation of $r$ in episode $t$), and horizon $h$ at the $t$-th episode, where the empirical $Q$-value is evaluated by the estimated transition kernels $\hat{p}_h^{\text{M},t}$ and reward function $\hat{r}_h^t$ at the $t$-th episode. We immediately have the following bound

$$\left| V_1^\star(s_1) - \widehat{V}_1^{\tau_1, \pi^\star}(s_1) \right| + \left| \widehat{V}_1^{\tau_1, \hat{\pi}^{\tau_1}}(s_1) - V_1^{\hat{\pi}^{\tau_1}}(s_1) \right| \leq \hat{e}_1^{t,\pi^*}(s_1, \pi^*(s_1); r^{\text{M}}) + \hat{e}_1^{t,\hat{\pi}^{\tau_1}}(s_1, \hat{\pi}^{\tau_1}(s_1); r^{\text{M}}).$$

Notice that here, our definition of $\hat{e}_h^{t,\pi}(s, a; r)$ incorporates random reward functions, and is different from that of Ménard et al. (2021). Next, we show the following uniform bound for $\hat{e}_1^{t,\pi}(s_1, \pi(s_1); r)$.

**Lemma 3.** *With probability at least $1 - \delta$, for any episode $t$, policy $\pi$, and reward function $r$ that is in $[0, 1]$,*

$$\hat{e}_1^{t,\pi}(s_1, \pi(s_1); r) \leq 4e\sqrt{\max_{a \in \mathcal{A}} W_1^t(s_1, a)} + \max_{a \in \mathcal{A}} W_1^t(s_1, a).$$

Recall $W_1^t(\cdot, \cdot)$ is defined by equation (5) and it is a function of $\delta$ (we omit the dependence of $\delta$ in $W$ for notation simplicity). With Lemma 3, we know that for our quantity of interest, we just need to bound $4e\sqrt{\max_{a \in \mathcal{A}} W_1^t(s_1, a)} + \max_{a \in \mathcal{A}} W_1^t(s_1, a)$ with the following lemma.

**Lemma 4.** *For $\epsilon > 0$ and $\delta > 0$, with probability at least $1 - \delta$, we have*

$$4e\sqrt{\max_{a \in \mathcal{A}} W_1^{\tau_1}(s_1, a)} + \max_{a \in \mathcal{A}} W_1^{\tau_1}(s_1, a) \leq \epsilon/2$$

*and the terminating time $\tau_1$ is bounded by*

$$\tau_1 \leq \frac{H^3 S A}{\varepsilon^2} (\log(4SAH/\delta) + S) C_1 + 1,$$

*where $C_1 = 9000e^6 \log^2 \left( e^{18} (\log(4HSAT/\delta) + S) \frac{H^4 SA}{\epsilon} \right)$.*

With Lemma 4, we know that with probability $1 - \delta$,

$$V_1^\star (s_1) - V_1^{\widehat{\pi}^{\tau_1}} (s_1) \leq \left| V_1^\star (s_1) - \widehat{V}_1^{\tau_1,\pi^\star} (s_1) \right| + \left| \widehat{V}_1^{\tau_1,\widehat{\pi}^{\tau_1}} (s_1) - V_1^{\widehat{\pi}^{\tau_1}} (s_1) \right|$$

$$\leq \hat{e}_h^{t,\pi^*} (s_1, \pi^*(s_1); r^{\mathrm{M}}) + \hat{e}_h^{t,\widehat{\pi}^{\tau_1}} (s_1, \widehat{\pi}^{\tau_1}(s_1); r^{\mathrm{M}})$$

$$\leq 2 \left( 4e \sqrt{\max_{a \in \mathcal{A}} W_1^t(s_1,a)} + \max_{a \in \mathcal{A}} W_1^t(s_1,a) \right) \leq \epsilon.$$

**Upper bounds of $\tau$ and $V_{1,\beta}^*(s_1) - V_{1,\beta}^{\widehat{\pi}_\beta^\tau}(s_1)$.**

To establish the result for $\tau$, we first scale the reward from $[0, H]$ to $[0, 1]$. From Algorithm 2 and Lemma 4, by setting the threshold to $\epsilon/(2H)$, we know that with probability $1 - \delta$, $\tau = \tilde{O}(H^3 S^2 A/(\epsilon^2/H^2)) = \tilde{O}(H^5 S^2 A/\epsilon^2)$. The rest is to show that we have for all $\beta \in (0, H)$,

$$V_{1,\beta}^*(s_1) - V_{1,\beta}^{\widehat{\pi}_\beta^\tau}(s_1) \leq \epsilon.$$

With an analysis similar to that of $\tau_1$, for any $\beta \in [0, H)$, when we scale back the reward to $[0, H]$ (notice the multiplier $H$ in the last inequality), we have

$$V_{1,\beta}^\star (s_1) - V_{1,\beta}^{\widehat{\pi}_\beta^\tau} (s_1) = V_{1,\beta}^\star (s_1) - \widehat{V}_{1,\beta}^{\tau,\pi_\beta^\star} (s_1) + \widehat{V}_{1,\beta}^{\tau,\pi_\beta^\star} (s_1) - \widehat{V}_{1,\beta}^{\tau,\widehat{\pi}_\beta^\tau} (s_1) + \widehat{V}_{1,\beta}^{\tau,\widehat{\pi}_\beta^\tau} (s_1) - V_{1,\beta}^{\widehat{\pi}_\beta^\tau} (s_1)$$

$$\leq \left| V_{1,\beta}^\star (s_1) - \widehat{V}_{1,\beta}^{\tau,\pi_\beta^\star} (s_1) \right| + \left| \widehat{V}_{1,\beta}^{\tau,\widehat{\pi}_\beta^\tau} (s_1) - V_{1,\beta}^{\widehat{\pi}_\beta^\tau} (s_1) \right|$$

$$\leq \hat{e}_1^{\tau,\pi_\beta^*} (s_1, \pi_\beta^*(s_1); r_\beta^{\mathrm{M}}) + \hat{e}_1^{\tau,\widehat{\pi}_\beta^\tau} (s_1, \hat{\pi}_\beta^\tau(s_1); r_\beta^{\mathrm{M}})$$

$$\leq 8e \sqrt{\max_{a \in \mathcal{A}} W_1^\tau(s_1,a)} + 2 \max_{a \in \mathcal{A}} W_1^\tau(s_1,a) \leq 2H\epsilon/(2H) = \epsilon.$$

To conclude the proof, we just have to notice that $8e \sqrt{\max_{a \in \mathcal{A}} W_1^\tau(s_1,a)} + 2 \max_{a \in \mathcal{A}} W_1^\tau(s_1,a)$ is a bound for any reward function $r_\beta^{\mathrm{M}}$ with $\beta \in [0, H)$.

**Proof for Corollary 1.**

Recall that we have the following CMDP and sample CMDP defined as

$$\max_\pi \ \mathbb{E}^\pi \left[ \sum_{h=1}^H r^{\mathrm{M}}(s_h, a_h) \right] \quad s.t. \ \mathbb{E}^\pi \left[ \sum_{h=1}^H \mathbb{I}\{a_h \neq \mathrm{defer}\} \right] \leq D, \quad (21)$$

$$\max_\pi \ \widehat{\mathbb{E}}^\pi \left[ \sum_{h=1}^H \hat{r}^{\mathrm{M},\tau}(s_h, a_h) \right] \quad s.t. \ \widehat{\mathbb{E}}^\pi \left[ \sum_{h=1}^H \mathbb{I}\{a_h \neq \mathrm{defer}\} \right] \leq D, \quad (22)$$

where $\widehat{\mathbb{E}}$ denotes that the transition kernel follows $\hat{p}^{\mathrm{M},\tau}$, and $\pi_D^*$ and $\hat{\pi}_D^\tau$ are the corresponding solutions for the above CMDP problems.

From the standard primal-dual theorem, there exists non-negative $\beta_D^*$, which is the optimal dual variable for the constraint $\mathbb{E}[\sum_{h=1}^H \mathbb{I}\{a_h \neq \mathrm{defer}\}] \leq D$, such that $\beta_D^*$ and the optimal policy for (21) solve the following saddle point problem

$$\min_{\beta \geq 0} \max_{\pi \in \Pi} \left( \mathbb{E}^\pi \left[ \sum_{h=1}^H r^{\mathrm{M}}(s_h, a_h) \right] + \beta \left( D - \mathbb{E}^\pi \left[ \sum_{h=1}^H \mathbb{I}\{a_h \neq \mathrm{defer}\} \right] \right) \right)$$

$$= \min_{\beta \geq 0} \max_{\pi \in \Pi} \left( V_\beta^\pi(s_1) + \beta D \right).$$

We observe that the optimal policy for (21) is the same as the optimal policy that maximize $V_{\beta_D^*}^\pi$. Therefore, adding penalty $\beta$ for advice and solving $V_\beta^*$ are equivalent to imposing constraint to the original MDP and solving the corresponding CMDP problem.

Now let us start the proof. For the proof of constraint violation, from the results for $\tau_1$, we can view $\mathbb{E}^\pi \left[ \sum_{h=1}^H \mathbb{I}\{a_h \neq \mathrm{defer}\} \right]$ and $\widehat{\mathbb{E}}^\pi \left[ \sum_{h=1}^H \mathbb{I}\{a_h \neq \mathrm{defer}\} \right]$ as value functions under transition

kernels $\hat{p}^{\mathrm{M},\tau}$ and $p^{\mathrm{M}}$. Therefore by knowing that

$$\left| \mathbb{E}^{\hat{\pi}_D^\tau} \left[ \sum_{h=1}^{H} \mathbb{I}\{a_h \neq \mathrm{defer}\} \right] - \hat{\mathbb{E}}^{\hat{\pi}_D^\tau} \left[ \sum_{h=1}^{H} \mathbb{I}\{a_h \neq \mathrm{defer}\} \right] \right| \leq \frac{\epsilon}{H}, \quad \text{and} \quad \hat{\mathbb{E}}^{\hat{\pi}_D^\tau} \left[ \sum_{h=1}^{H} \mathbb{I}\{a_h \neq \mathrm{defer}\} \right] \leq D,$$

we have

$$\mathbb{E}^{\hat{\pi}_D^\tau} \left[ \sum_{h=1}^{H} \mathbb{I}\{a_h \neq \mathrm{defer}\} \right] \leq D + \epsilon.$$

To show the $\epsilon$-optimal property of the objective function, by observing that $\mathbb{E}^{\pi} \left[ \sum_{h=1}^{H} r^{\mathrm{M}}(s_h, a_h) \right] = V_1^{\pi}(s_1)$, we can get the following decomposition

$$V_1^{\pi_D^*}(s_1) - V_1^{\hat{\pi}_D^\tau}(s_1) = V_1^{\pi_D^*}(s_1) - \widehat{V}_1^{\pi_D^*}(s_1) + \widehat{V}_1^{\pi_D^*}(s_1) - \widehat{V}_1^{\hat{\pi}_D^\tau}(s_1) + \widehat{V}_1^{\hat{\pi}_D^\tau}(s_1) - V_1^{\hat{\pi}_D^\tau}(s_1)$$

$$\leq \left| V_1^{\pi_D^*}(s_1) - \widehat{V}_1^{\pi_D^*}(s_1) \right| + \left| \widehat{V}_1^{\hat{\pi}_D^\tau}(s_1) - V_1^{\hat{\pi}_D^\tau}(s_1) \right| + \widehat{V}_1^{\pi_D^*}(s_1) - \widehat{V}_1^{\hat{\pi}_D^\tau}(s_1)$$

$$\leq \frac{2\epsilon}{H} + \widehat{V}_1^{\pi_D^*}(s_1) - \widehat{V}_1^{\hat{\pi}_D^\tau}(s_1),$$

where in the last inequality, we have the bound $\epsilon/H$ because the reward is in the scale $(0,1)$ and the terminating time is $\tau$. For the term $\widehat{V}_1^{\pi_D^*}(s_1) - \widehat{V}_1^{\hat{\pi}_D^\tau}(s_1)$, we note that from the primal-dual property, based on the primal problem (22), there exists $\hat{\beta}_D \in [0, H)$ such that $\hat{\pi}_D^\tau = \arg\max_{\pi \in \Pi} \hat{V}_{\hat{\beta}_D}^{\pi}(s)$. Therefore, we know that

$$\hat{V}_{\hat{\beta}_D}^{\hat{\pi}_D^\tau}(s_1) \geq \hat{V}_{\hat{\beta}_D}^{\pi_D^*}(s_1),$$

which implies

$$\widehat{V}_1^{\pi_D^*}(s_1) - \widehat{V}_1^{\hat{\pi}_D^\tau}(s_1) \leq \hat{\beta}_D \left( \hat{\mathbb{E}}^{\pi_D^*} \left[ \sum_{h=1}^{H} \mathbb{I}\{a_h \neq \mathrm{defer}\} \right] - \hat{\mathbb{E}}^{\hat{\pi}_D^\tau} \left[ \sum_{h=1}^{H} \mathbb{I}\{a_h \neq \mathrm{defer}\} \right] \right).$$

Next, we discuss different cases on $\hat{\beta}_D$. If $\hat{\beta}_D = 0$, this means that $\hat{\mathbb{E}}^{\hat{\pi}_D^\tau} \left[ \sum_{h=1}^{H} \mathbb{I}\{a_h \neq \mathrm{defer}\} \right] < D$ and the constrained optimization problem can be treated as an unconstrained one. Therefore we have $\widehat{V}_1^{\hat{\pi}_D^\tau}(s_1)$ being the optimal solution for the unconstrained problem, and $\widehat{V}_1^{\pi_D^*}(s_1) - \widehat{V}_1^{\hat{\pi}_D^\tau}(s_1) \leq 0$. For the case where $\hat{\beta}_D \in (0, H)$, we have $\hat{\mathbb{E}}^{\hat{\pi}_D^\tau} \left[ \sum_{h=1}^{H} \mathbb{I}\{a_h \neq \mathrm{defer}\} \right] = D$ and

$$\widehat{V}_1^{\pi_D^*}(s_1) - \widehat{V}_1^{\hat{\pi}_D^\tau}(s_1) \leq \hat{\beta}_D \left( \hat{\mathbb{E}}^{\pi_D^*} \left[ \sum_{h=1}^{H} \mathbb{I}\{a_h \neq \mathrm{defer}\} \right] - D \right).$$

By viewing $\hat{\mathbb{E}}^{\pi_D^*} \left[ \sum_{h=1}^{H} \mathbb{I}\{a_h \neq \mathrm{defer}\} \right]$ as value function, from the property of $\tau$ and the original CMDP (21) we know that

$$\left| \hat{\mathbb{E}}^{\pi_D^*} \left[ \sum_{h=1}^{H} \mathbb{I}\{a_h \neq \mathrm{defer}\} \right] - \mathbb{E}^{\pi_D^*} \left[ \sum_{h=1}^{H} \mathbb{I}\{a_h \neq \mathrm{defer}\} \right] \right| \leq \frac{\epsilon}{H}, \quad \text{and} \quad \mathbb{E}^{\pi_D^*} \left[ \sum_{h=1}^{H} \mathbb{I}\{a_h \neq \mathrm{defer}\} \right] \leq D.$$

Therefore, we have $\hat{\mathbb{E}}^{\pi_D^*} \left[ \sum_{h=1}^{H} \mathbb{I}\{a_h \neq \mathrm{defer}\} \right] \leq D + \epsilon/H$, and $\widehat{V}_1^{\pi_D^*}(s_1) - \widehat{V}_1^{\hat{\pi}_D^\tau}(s_1) \leq H(\epsilon/H) = \epsilon$. Finally, we conclude the proof by observing that (using the convention $H > 1$)

$$V_1^{\pi_D^*}(s_1) - V_1^{\hat{\pi}_D^\tau}(s_1) \leq \frac{2\epsilon}{H} + \widehat{V}_1^{\pi_D^*}(s_1) - \widehat{V}_1^{\hat{\pi}_D^\tau}(s_1) \leq \frac{2\epsilon}{H} + \epsilon \leq 2\epsilon.$$

### A.3.3   PROOF OF ESSENTIAL LEMMAS FOR THEOREM 2

In this section, we list the proof of lemmas for Theorem 2. First, we need to introduce the high-probability event to characterize the high-probability bound.

Denote $p_h^{M,\pi}(s,a)$ the probability that the pair $(s,a)$ is visited under the model's transition kernel at time $t$ following policy $\pi$, and $\pi^t$ the policy of Algorithm 2 at episode $t$. We define the pseudo-counts to be

$$\bar{n}_h^t(s,a) = \sum_{l=1}^{t} p_h^{M,\pi^l}(s,a).$$

We define the following events that are favorable: $\mathcal{E}$, the event where the empirical transition probabilities are close to the true ones; $\mathcal{E}^{\text{cnt}}$, the event where the counts are close to pseudo-counts, their expectations; and $\mathcal{E}^r$, the event where the estimation of reward function is close to the expected reward function within sufficient large episodes. More specifically

$$\mathcal{E} \triangleq \left\{ \forall t \in \mathbb{N}, \forall h \in [H], \forall (s,a) \in \mathcal{S} \times \mathcal{A} : \text{KL}\left(\widehat{p}_h^{M,t}(\cdot \mid s,a), p_h^{M}(\cdot \mid s,a)\right) \leq \frac{\beta\left(n_h^t(s,a),\delta\right)}{n_h^t(s,a)} \right\},$$

$$\mathcal{E}^{\text{cnt}} \triangleq \left\{ \forall t \in \mathbb{N}, \forall h \in [H], \forall (s,a) \in \mathcal{S} \times \mathcal{A} : n_h^t(s,a) \geq \frac{1}{2}\bar{n}_h^t(s,a) - \beta^{\text{cnt}}(\delta) \right\},$$

$$\mathcal{E}^r \triangleq \left\{ \forall t \leq 4^{10}e^{12}S^{11}A^{11}H^{11}/(\epsilon^{12}\delta^{12}), \forall h \in [H], \forall (s,a) \in \mathcal{S} \times \mathcal{A} : |r_h^{M}(s,a) - \hat{r}_h^{M}(s,a)| \leq \sqrt{\frac{\beta^r\left(n_h^t(s,a)\right)}{n_h^t(s,a)}} \right\}$$

$$(23)$$

where $\text{KL}(\cdot,\cdot)$ is the KL-divergence of two distributions. We define the $\beta$ functions and show the high probability events in the following lemma

**Lemma 5.** *For the following choices of functions $\beta$,*

$$\phi(n,\delta) = 6\log(4eSAH/(\epsilon\delta)) + S\log(8e(n+1)),$$
$$\beta^{\text{cnt}}(\delta) = 6\log(4eSAH/(\epsilon\delta)),$$
$$\beta^r(\delta) = 6\log(4eSAH/(\epsilon\delta)),$$

*it holds that*

$$P(\mathcal{E}) \geq 1 - \delta/3, \quad P\left(\mathcal{E}^{\text{cnt}}\right) \geq 1 - \delta/3, \text{ and } P\left(\mathcal{E}^{\text{r}}\right) \geq 1 - \delta/3$$

With Lemma 5 that characterize the high probability event $\mathcal{E} \cap \mathcal{E}^{cnt} \cap \mathcal{E}^r$, we are able to prove lemma 3 and 4.

*Proof of Lemma 3.* The proof is similar to the proof of Lemma 1 in Ménard et al. (2021), and the difference is that we further analyze the estimation error for the rewards while they assume that the reward function is known and deterministic. From lemma 5, the event $\mathcal{E} \cap \mathcal{E}^r$ has probability at least $1 - \delta$, and in this proof every computation is carried out assuming under $\mathcal{E} \cap \mathcal{E}^r$. For any policy $\pi$, we have

$$
\begin{aligned}
\hat{e}_h^{t,\pi}(s,a;r) =& |\hat{Q}_h^{t,\pi}(s,a;r) - Q_h^{\pi}(s,a;r)| \\
\leq& |r_h^{M}(s,a) - \hat{r}_h^{M,t}(s,a)| + \left| \sum_{s' \in \mathcal{S}} p_h^{M}(s'|s,a) Q_{h+1}^{\pi}(s',\pi_{h+1}(s')) - \sum_{s' \in \mathcal{S}} \hat{p}_h^{M,t}(s'|s,a) \hat{Q}_{h+1}^{t,\pi}(s',\pi_{h+1}(s')) \right|, \\
\leq& |r_h^{M}(s,a) - \hat{r}_h^{M,t}(s,a)| + \left| \sum_{s' \in \mathcal{S}} (p_h^{M}(s'|s,a) - \hat{p}_h^{M,t}(s'|s,a)) Q_{h+1}^{\pi}(s',\pi_{h+1}(s')) \right| \quad (24) \\
& + \left| \sum_{s' \in \mathcal{S}} \hat{p}_h^{M,t}(s'|s,a)(Q_{h+1}^{\pi}(s',\pi_{h+1}(s')) - \hat{Q}_{h+1}^{t,\pi}(s',\pi_{h+1}(s'))) \right| \\
\leq& |r_h^{M}(s,a) - \hat{r}_h^{M,t}(s,a)| + \sum_{s' \in \mathcal{S}} \left| p_h^{M}(s'|s,a) - \hat{p}_h^{M,t}(s'|s,a) \right| Q_{h+1}^{\pi}(s',\pi_{h+1}(s')) \\
& + \sum_{s' \in \mathcal{S}} \hat{p}_h^{M,t}(s'|s,a) \left| Q_{h+1}^{\pi}(s',\pi_{h+1}(s')) - \hat{Q}_{h+1}^{t,\pi}(s',\pi_{h+1}(s')) \right|,
\end{aligned}
$$

$$(25)$$

where the first step comes from the definition of $\hat{e}_h^{t,\pi}(s,a;r)$, the second step comes from the recursive definition of the $Q$-values (20), and the last step comes from the triangle inequality for the absolute value. Next, we will apply empirical Bernstein's inequality to the terms above. Specifically,

$$\leq |r_h^{\mathtt{M}}(s,a) - \hat{r}_h^{\mathtt{M},t}(s,a)| + 3\sqrt{\frac{\mathrm{Var}_{\hat{p}_h^{\mathtt{M},t}(\cdot|s,a)}(\hat{V}_{h+1}^{t,\pi})}{H^2}\left(\frac{H^2\beta\left(n_h^t(s,a),\delta\right)}{n_h^t(s,a)} \wedge 1\right)} + 15H^2\frac{\phi(n_h^t(s,a),\delta)}{n_h^t(s,a)}$$

$$+\left(1+\frac{1}{H}\right)\sum_{s'\in\mathcal{S}}\hat{p}_h^{\mathtt{M},t}(s'|s,a)\hat{e}_{h+1}^{t,\pi}(s',\pi_{h+1}(s');r).$$

$$\leq \sqrt{\frac{\beta^r(n_h^t(s,a))}{n_h^t(s,a)}} + 3\sqrt{\frac{\mathrm{Var}_{\hat{p}_h^{\mathtt{M},t}(\cdot|s,a)}(\hat{V}_{h+1}^{t,\pi})}{H^2}\left(\frac{H^2\beta\left(n_h^t(s,a),\delta\right)}{n_h^t(s,a)} \wedge 1\right)} + 15H^2\frac{\phi(n_h^t(s,a),\delta)}{n_h^t(s,a)}$$

$$+\left(1+\frac{1}{H}\right)\sum_{s'\in\mathcal{S}}\hat{p}_h^{\mathtt{M},t}(s'|s,a)\hat{e}_{h+1}^{t,\pi}(s',\pi_{h+1}(s');r), \tag{26}$$

where the first inequality follows in Ménard et al. (2021) page 8 (the readers can also see there how the high probability event $\mathcal{E}$ is used in page 8 and Lemma 10 there), and the second comes from $\mathcal{E}^r$. Then, observe that $\beta^r(n) \leq \phi(n,\delta)$, and we have

$$\sqrt{\frac{\phi(n_h^t(s,a),\delta)}{n_h^t(s,a)}} = \sqrt{\frac{1}{H^2}\frac{H^2\phi(n_h^t(s,a),\delta)}{n_h^t(s,a)}} \leq \sqrt{\frac{1}{H^2}\left(\frac{H^2\beta\left(n_h^t(s,a),\delta\right)}{n_h^t(s,a)} \wedge 1\right)} + \frac{1}{H}\frac{H^2\phi(n_h^t(s,a),\delta)}{n_h^t(s,a)},$$

for the reason that $\sqrt{x} \leq x$ if $x \geq 1$. Therefore, we have

$$\hat{e}_h^{t,\pi}(s,a;r) \leq \left(\frac{1}{H} + 3\sqrt{\frac{\mathrm{Var}_{\hat{p}_h^{\mathtt{M},t}(\cdot|s,a)}(\hat{V}_{h+1}^{t,\pi})}{H^2}}\right)\sqrt{\left(\frac{H^2\beta\left(n_h^t(s,a),\delta\right)}{n_h^t(s,a)} \wedge 1\right)} + 16H^2\frac{\phi(n_h^t(s,a),\delta)}{n_h^t(s,a)}$$

$$+\left(1+\frac{1}{H}\right)\sum_{s'\in\mathcal{S}}\hat{p}_h^{\mathtt{M},t}(s'|s,a)\hat{e}_{h+1}^{t,\pi}(s',\pi_{h+1}(s');r). \tag{27}$$

Then, we can recursively define an **upper bound** for $\hat{e}_h^{t,\pi}(s,a;r)$ based on (27) as follows:

$$Z_h^{t,\pi}(s,a;r) = \min\left\{H, \left(\frac{1}{H} + 3\sqrt{\frac{\mathrm{Var}_{\hat{p}_h^{\mathtt{M},t}(\cdot|s,a)}(\hat{V}_{h+1}^{t,\pi})}{H^2}}\right)\sqrt{\left(\frac{H^2\beta\left(n_h^t(s,a),\delta\right)}{n_h^t(s,a)} \wedge 1\right)}\right.$$

$$\left.+16H^2\frac{\phi(n_h^t(s,a),\delta)}{n_h^t(s,a)} + \left(1+\frac{1}{H}\right)\sum_{s'\in\mathcal{S}}\hat{p}_h^{\mathtt{M},t}(s'|s,a)Z_{h+1}^{t,\pi}(s',\pi_{h+1}(s');r)\right\}.$$

Then, consider the following two sequences

$$Y_h^{t,\pi}(s,a;r) = \left(\frac{1}{H} + 3\sqrt{\frac{\mathrm{Var}_{\hat{p}_h^{\mathtt{M},t}(\cdot|s,a)}(\hat{V}_{h+1}^{t,\pi})}{H^2}}\right)\sqrt{\left(\frac{H^2\beta\left(n_h^t(s,a),\delta\right)}{n_h^t(s,a)} \wedge 1\right)}$$

$$+\left(1+\frac{1}{H}\right)\sum_{s'\in\mathcal{S}}\hat{p}_h^{\mathtt{M},t}(s'|s,a)Y_{h+1}^{t,\pi}(s',\pi_{h+1}(s');r),$$

$$W_h^{t,\pi}(s,a;r) = \min\left\{H, 16H^2\frac{\phi(n_h^t(s,a),\delta)}{n_h^t(s,a)}\left(1+\frac{1}{H}\right)\sum_{s'\in\mathcal{S}}\hat{p}_h^{\mathtt{M},t}(s'|s,a)Z_{h+1}^{t,\pi}(s',\pi_{h+1}(s');r)\right\}.$$

We can prove by induction that for all $h,s,a$,

$$\hat{e}_h^{t,\pi}(s,a;r) \leq Z_h^{t,\pi}(s,a;r) \leq Y_h^{t,\pi}(s,a;r) + W_h^{t,\pi}(s,a).$$

Therefore, to bound $\hat{e}_1^{t,\pi}(s_1,\pi(s_1);r)$, it suffices to bound $Y_h^{t,\pi}(s_1,\pi(s_1);r) + W_h^{t,\pi}(s_1,\pi(s_1))$. Denote $\hat{p}_h^{\mathtt{M},t,\pi}(s,a)$ the probability that the pair $(s,a)$ is visited under the estimated transition kernel following policy $\pi$, we have

$$Y_1^{t,\pi}(s_1, \pi(s_1); r)$$

$$= \sum_{s,a} \sum_{h=1}^{H} \widehat{p}_h^{\mathtt{M},t,\pi}(s,a) \left(1 + \frac{1}{H}\right)^{h-1} \left(\frac{1}{H} + 3\sqrt{\frac{\mathrm{Var}_{\widehat{p}_h^{\mathtt{M},t}(\cdot|s,a)}(\hat{V}_{h+1}^{t,\pi})}{H^2}}\right) \sqrt{\left(\frac{H^2 \beta\left(n_h^t(s,a),\delta\right)}{n_h^t(s,a)} \wedge 1\right)}$$

$$\leq 3e \sqrt{\sum_{s,a} \sum_{h=1}^{H} \widehat{p}_h^{\mathtt{M},t,\pi}(s,a) \frac{\mathrm{Var}_{\widehat{p}_h^{\mathtt{M},t}(\cdot|s,a)}(\hat{V}_{h+1}^{t,\pi})}{H^2}} \sqrt{\sum_{s,a} \widehat{p}_{h=1}^{t,\pi}(s,a) \left(\frac{H^2 \beta\left(n_h^t(s,a),\delta\right)}{n_h^t(s,a)} \wedge 1\right)}$$

$$+ e \sqrt{\sum_{s,a} \sum_{h=1}^{H} \widehat{p}_h^{\mathtt{M},t,\pi}(s,a) \frac{1}{H^2}} \sqrt{\sum_{s,a} \widehat{p}_{h=1}^{t,\pi}(s,a) \left(\frac{H^2 \beta\left(n_h^t(s,a),\delta\right)}{n_h^t(s,a)} \wedge 1\right)}$$

$$\leq \left(3e \sqrt{\frac{1}{H^2} \mathbb{E}_{\pi,\widehat{p}_h^{\mathtt{M},t}} \left[\left(\sum_{h=1}^{H} r_h\left(s_h, a_h\right) - \hat{V}_1^{\pi}\left(s_1; r\right)\right)^2\right]} + \frac{e}{\sqrt{H}}\right) \sqrt{\sum_{s,a} \sum_{h=1}^{H} \widehat{p}_h^{\mathtt{M},t,\pi}(s,a) \left(\frac{H^2 \beta\left(n_h^t(s,a),\delta\right)}{n_h^t(s,a)} \wedge 1\right)}$$

$$\leq 4e \sqrt{\sum_{s,a} \sum_{h=1}^{H} \widehat{p}_h^{\mathtt{M},t,\pi}(s,a) \left(\frac{H^2 \beta\left(n_h^t(s,a),\delta\right)}{n_h^t(s,a)} \wedge 1\right)} \leq 4e \sqrt{W_1^{t,\pi}\left(s_1, \pi(s_1)\right)},$$

Where the second inequality comes from the law of total variance (Ménard et al. (2021) Lemma 7), and the last inequality comes from page 24 (Step 3) in Ménard et al. (2021). Therefore, we have

$$\hat{e}_h^{t,\pi}(s_1, \pi(s_1); r) \leq 4e \sqrt{\max_{a \in \mathcal{A}} W_1^t(s_1, a)} + \max_{a \in \mathcal{A}} W_1^t(s_1, a).$$

$$\square$$

*Proof of Lemma 4.* We first provide an upper bound on $W_h^t(s,a)$ for all $(s,a,h)$ and $t$. By definition (5), if $n_h^t(s,a) > 0$, we have

$$W_h^t(s,a) \leq 16H^2 \frac{\beta\left(n_h^t(s,a),\delta\right)}{n_h^t(s,a)} + \left(1 + \frac{1}{H}\right) \sum_{s'} \widehat{p}_h^{\mathtt{M},t}\left(s' \mid s,a\right) \max_{a'} W_{h+1}^t\left(s', a'\right)$$

$$= 16H^2 \frac{\beta\left(n_h^t(s,a),\delta\right)}{n_h^t(s,a)} + \left(1 + \frac{1}{H}\right) \sum_{s' \in \mathcal{S}} \left(\widehat{p}_h^{\mathtt{M},t}(s'|s,a) - p_h^{\mathtt{M}}(s'|s,a)\right) W_{h+1}^t(s', \pi_{h+1}^{t+1}(s'))$$

$$+ \left(1 + \frac{1}{H}\right) \sum_{s' \in \mathcal{S}} p_h^{\mathtt{M}}(s'|s,a) W_{h+1}^t(s', \pi_{h+1}^{t+1}(s'))$$

From Lemma 10 in Ménard et al. (2021) and the Bernstein inequality we get (see page 25 in Ménard et al. (2021) for more details)

$$W_h^t(s,a) \leq 22H^2 \left(\frac{\beta\left(n_h^t(s,a),\delta\right)}{n_h^t(s,a)} \wedge 1\right) + \left(1 + \frac{3}{H}\right) \sum_{s' \in \mathcal{S}} p_h(s'|s,a) W_{h+1}^t(s, \pi_{h+1}^{t+1}(s))$$

Unfolding the above equation and using $(1 + 3/H)^H \leq e^3$ we have

$$W_1^t\left(s_1, \pi_1^{t+1}(s_1)\right) \leq 22e^3 H^2 \sum_{h=1}^{H} \sum_{s,a} p_h^{\mathtt{M},t+1}(s,a) \left(\frac{\beta\left(n_h^t(s,a),\delta\right)}{n_h^t(s,a)} \wedge 1\right)$$

In this proof, we choose the high probability event to be $\mathcal{E}^{cnt}$, under which we have (see Ménard et al. (2021) lemma 8)

$$W_1^t\left(s_1, \pi_1^{t+1}(s_1)\right) \leq 88e^3 H^2 \sum_{h=1}^{H} \sum_{s,a} p_h^{\mathtt{M},t+1}(s,a) \frac{\beta\left(\bar{n}_h^t(s,a),\delta\right)}{\bar{n}_h^t(s,a) \vee 1}, \tag{28}$$

where we recall that $\bar{n}_h^t(s,a) = \sum_{l=1}^t p_h^{\mathtt{M},\pi^l}(s,a)$ is the pseudo-count.

Next, we are going to sum the above inequality over $t \leq T$ for $T < \tau$. Due to the stopping rule, we have

$$\varepsilon \leq 4e\sqrt{W_1^t\left(s_1, \pi_1^{t+1}(s_1)\right)} + W_1^t\left(s_1, \pi_1^{t+1}(s_1)\right).$$

Summing the over the above inequalities for $0 \leq t \leq T$, followed by Cauchy-Schwarz inequality, we have

$$(T+1)\varepsilon \leq \sum_{t=0}^T \left(4e\sqrt{W_1^t\left(s_1, \pi_1^{t+1}(s_1)\right)} + W_1^t\left(s_1, \pi_1^{t+1}(s_1)\right)\right)$$

$$\leq 4e\sqrt{(T+1)\sum_{t=0}^T W_1^t\left(s_1, \pi_1^{t+1}(s_1)\right)} + \sum_{t=0}^T W_1^t\left(s_1, \pi_1^{t+1}(s_1)\right).$$

Next, from (28), the property that $\phi(\cdot, \delta)$ is increasing and Lemma 9 in Ménard et al. (2021), we have

$$\sum_{t=0}^T W_1^t\left(s_1, \pi_1^{t+1}(s_1)\right) \leq 88e^3 H^2 \sum_{t=0}^T \sum_{h=1}^H \sum_{s,a} p_h^{\mathtt{M},t+1}(s,a) \frac{\beta\left(\bar{n}_h^t(s,a), \delta\right)}{\bar{n}_h^t(s,a) \vee 1}$$

$$\leq 88e^3 H^2 \phi(T, \delta) \sum_{t=0}^T \sum_{h=1}^H \sum_{s,a} p_h^{\mathtt{M},t+1}(s,a) \frac{1}{\bar{n}_h^t(s,a) \vee 1}$$

$$= 88e^3 H^2 \phi(T, \delta) \sum_{h=1}^H \sum_{s,a} \sum_{t=0}^T \frac{\bar{n}_h^{t+1}(s,a) - \bar{n}_h^t(s,a)}{\bar{n}_h^t(s,a) \vee 1}$$

$$\leq 352e^3 H^3 SA \log(T+2)\phi(T, \delta)$$

Therefore, we have

$$(T+1)\varepsilon \leq 76e^3\sqrt{(T+1)H^3 SA \log(T+2)\phi(T, \delta)} + 352e^3 H^3 SA \log(T+2)\phi(T, \delta)$$

Lastly, using Lemma 13 in Ménard et al. (2021) we have

$$\tau \leq \frac{H^3 SA}{\varepsilon^2}(\log(4SAH/\delta) + S)C_1 + 1$$

where $C_1 = 9000e^6 \log^2\left(e^{18}\left(\log(4HSAT/\delta) + S\right)\frac{H^4 SA}{\epsilon}\right)$.

$\square$

*Proof of Lemma 5.* To prove the probability bound for the first two sets $\mathcal{E}$ and $\mathcal{E}^{cnt}$, we refer to Lemma 3 in Ménard et al. (2021). To show the probability bound for $\mathcal{E}^r$, by Hoeffding's inequality, we have with probability at least $1 - \frac{\delta}{4^{11}e^{12}S^{12}A^{12}H^{12}}$

$$|r_h^{\mathtt{M}}(s,a) - \hat{r}_h^{\mathtt{M}}(s,a)| \leq \frac{\beta^r\left(n_h^t(s,a)\right)}{\sqrt{n_h^t(s,a)}} \tag{29}$$

for any fixed episode $t$, step $h$, and state-action pair $(s,a) \in \mathcal{S} \times \mathcal{A}$. Then, taking a union bound we have (29) holds for all state-action pairs, and $t \leq 4^{10}e^{12}S^{11}A^{11}H^{11}/(\epsilon^{12}\delta^{12})$ with probability no less than $1 - \frac{\delta}{3}$.

We remark here that although we have this episode's upper bound, our algorithm will stop before reaching this upper bound. Specifically, by calculations, we have for all $H, A, S \geq 1$

$$\frac{4^{10}e^{12}S^{11}A^{11}H^{11}}{\epsilon^{12}\delta^{12}} \geq 1048576e^{12}\frac{S^{11}A^{11}H^{11}}{\epsilon^{12}\delta^{12}}$$

$$\geq \left(36 \cdot 10^4 e^6 + 32 \cdot 10^4 e^8\right)\frac{H^7 S^4 A^4}{\epsilon^4 \delta^2}$$

$$\geq C_1 \frac{H^3 SA}{\epsilon^2} \cdot \frac{4SAH}{\delta},$$

which is larger than the bound in Theorem 2. Thus, our choice of $\beta^r$ guarantees that with high probability, the algorithm will find an $\epsilon$-optimal solution before reaching the maximum number of episodes. $\square$

## B  Supplementary Materials for Experiment

### B.1  Flappy Bird Experiment

**More details for Figure 2.**

We implement three algorithms in Figure 2. For each algorithm, due to the computation efficiency, the policy will be updated every 5000 episode in Figure 2a (Policy Greedy) and 1000 episode in Figure 2b (Policy Safe), and then apply the updated policies for the next 1000/5000 episode respectively. All three algorithms are trained by $8 \times 10^5$ for Policy Greedy and $2 \times 10^5$ for Policy Safe. All experiments are repeated $5$ times with the mean of results being reported. More details about implementations are as follows:

- RFE-ADvice (FRE-AD): Based on the estimated transition kernel and reward from Algorithm 2. At episode $t$, the algorithm uses planning oracles to obtain the optimal policy $\hat{\pi}^t$ for the MDP $\{\mathcal{S}, \bar{\mathcal{A}}, H, \hat{p}^{\mathrm{M},t}, r\}$. Note the rewards are known, we replace the reward's estimators $\hat{r}$ used in Algorithm 2 by the true reward $r$ of the environment. For the parameter of the algorithm, we set $\delta = 0.1$ and we do not set the convergence checking step so no need to input $\epsilon$. We also use $0.1 \cdot \beta$ instead of $\beta$ to control the "exploration" bonus (note $\beta$ function measures the uncertainty of the current state). In Figure 2a, 2b, we evaluate the value gap between the (estimated) optimal policy $\hat{\pi}^t$ and the optimal policy in every 1000 episode and use the cumulative sum of $1000 \times$ value gap as the "regret" of RFE-AD.

- UCB-ADherence (UCB-AD): This algorithm is the Algorithm 1 that outputs policy at different episode $t$ based on the upper confidence bound of the adherence level $\theta$. When implementing, we use

$$C(\theta, n, T, \delta) = 0.4 * \sqrt{\frac{2 \log(n)}{n}},$$

  as the function to measure uncertainty. Three subfigures of Figure 2 show the (cumulative) regret of UCB-AD. To test the robustness of UCB-AD, Figure 2c "zooms in" its regret under different values of $\theta$ for both policies.

- EULER: The algorithm is mainly based on Zanette and Brunskill (2019) with two differences. (1) Since the reward is known, we replace all estimates of reward $r$ by their true values (and set the uncertainty bonus as 0) in the algorithm; (2) The algorithm will fail when the unreachable states, where can not be arrived with probability 1 in the MDP, are not explicitly revealed. Since such states can not be known because of the unknown policy of humans, we feed additional $3 \times 10^5$ episodes for both policies as the exploration period (and thus the total number of training episodes is $11 \times 10^5$ and $5 \times 10^5$ for EULER), after which we set the states with $0$ observation/count as unreachable states. We further select the parameter $\delta = 0.1$ used in EULER. In Figure 2, we show the (cumulative) regret only after the exploration period ($3 \times 10^5$ episode).

**More details for Figure 3a.**

We implement RFE-$\beta$ in Figure 3a, which is based on the estimated transition kernel and reward from Algorithm 2 at episode $t$, the algorithm uses planning oracles to obtain the optimal policy $\{\mathcal{S}, \bar{\mathcal{A}}, H, \hat{p}^{\mathrm{M},t}, r_\beta\}$ with $\beta \in \{0, 0.2, 0.4\}$, and all the other settings are same as Figure 2. Figure 5 shows the value gaps of RFE-$\beta$, where for completeness we also include the copy of 3a.

**More details for Figure 3b, 3c.**

**Environment.** We change our environment to illustrate the challenge with advice budget, which is shown in Figure 6a. We should note to achieve the plotted advice policy, which is optimal without advice budget constraint for Policy Greedy, we should at least advise twice to make the bird go through the wall for two stars instead of getting the star at the beginning and hitting the wall. Thus, setting the advice budget as 1 will make both the policy itself and the learning process harder than with enough advice budget.

**Algorithm Implementation.** We implement two algorithms for this environment. For each algorithm, due to the computation efficiency, the policy will be updated every $50$ episodes, and then apply the

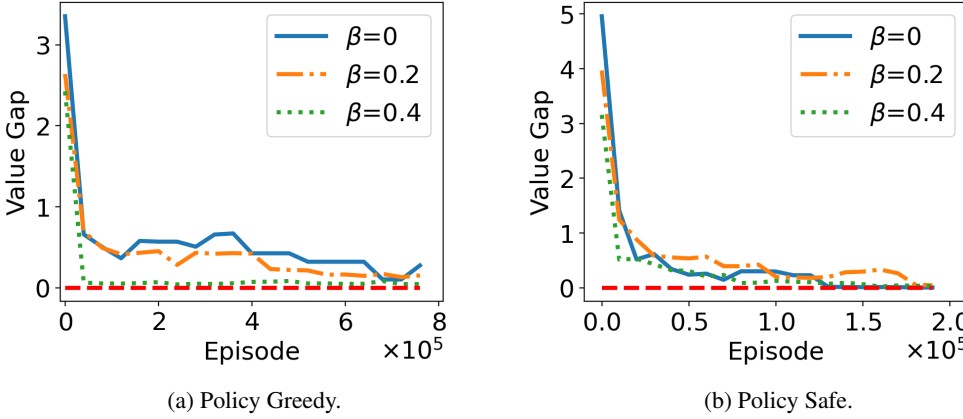

(a) Policy Greedy.

(b) Policy Safe.

Figure 5: Value Gaps of RFE-$\beta$.

updated policies for the next 50 episode. For Figure 3b and 6b, we evaluate both algorithms for every 50 episodes to compute the value gap and advice count gap. Both algorithms are trained by 1500 episode and all experiments are repeated 5 times with the mean of the results being reported. Details about implementations are as follows:

- RFE-CMDP: The algorithm is based on the estimated transition kernel and reward from Algorithm 2 at episode $t$ and it uses planning oracles to obtain the optimal policy $\hat{\pi}_D^t$ for the CMDP (7). For the algorithm's parameters, we set them as the same values as in 2.

- UC-CFH: The algorithm is based on Kalagarla et al. (2021). We set its parameter with $\epsilon = 1$ and $\delta = 0.5$ in this experiment.

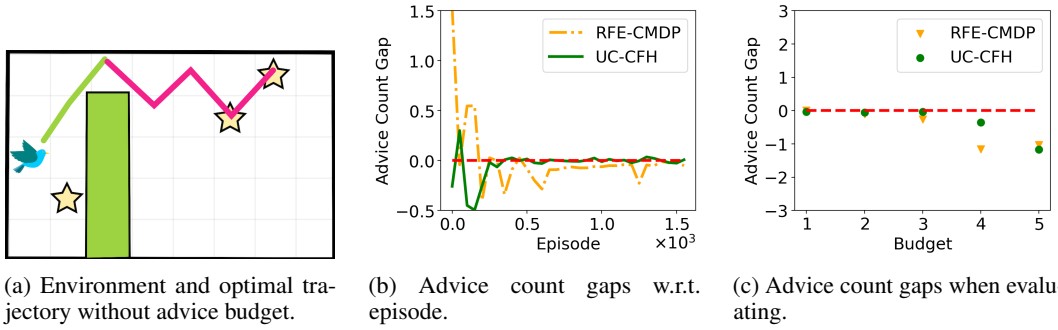

(a) Environment and optimal trajectory without advice budget.

(b) Advice count gaps w.r.t. episode.

(c) Advice count gaps when evaluating.

Figure 6: Additional results for RFE-CMDP. (a) shows the environment for testing the algorithms and one trajectory of optimal advice policy with advising twice: the red line means the machine defers and the green line means the machine advises. (b) and (c) show the budget violations through the advice count gap, which is computed by using the advice budget minus expected advising times of the policy, w.r.t. training episode and when evaluating respectively.

## B.2 CAR DRIVING ENVIRONMENT

In our car driving environment, we have three lanes and a horizon of 10. Each cell within the environment can either be empty, contain a stone, or have a car present. The types of cells follow independent and identically distributed (i.i.d.) distributions, which will be specified later. The objective for the driver is to maximize the distance covered by the car. However, colliding with another car or reaching the boundaries results in the destruction of the car, terminating the episode. Encountering a stone can also cause minor damage to the car, but the car will still be operating. The driver's goal is to drive on the empty road and avoid any obstacles. The experiment is adapted from Meresht et al. (2020).

**Car Driving MDP.**

We assume the machine is myopic up to two rows: It can observe the next two rows at most. Thus, the state space can be defined by the 9 cells' types in the current three rows with a total of $3^9 = 19683$ states. The environment can be represented by Figure 7a.

The action space is $\mathcal{A} = \{\text{Left, Straight, Right}\}$. The car will always keep moving to the next row unless it hits another car or the boundary. Further, the "Left" action will move the car to the left of the current lane, the "Right" action will move to the right, and the "Straight" action will keep the car in its current lane.

The car will get a reward of $1$ when the current cell where it is located is empty, a reward of $0.5$ when it has a stone, and a reward $0$ when it has a car or is out of the boundary (also, the environment will be terminated). The cell's type is sampled by $(0.4, 0.3, 0.3)$ for the empty, stone, and car respectively.

**Human behavior: $\pi^{\text{H}}$ and $\theta$.**

We consider a myopic driver who is only aware of other cars in the next row. Therefore, the driver's policy is to avoid the cars in the next row, and if there are multiple equivalent actions it will select them randomly with an equal probability. We set the adherence level $\theta$ as the follows

$$\theta(a, s) = \begin{cases} 0.9 & \text{if } a = \text{Straight,} \\ 0.7 & \text{otherwise.} \end{cases}$$

Thus, the driver will adhere to the advice with probability $0.9$ if the advice is "Straight" and otherwise $0.7$, and the intuition is that the driver wants to avoid changing the lane too often.

**Experiment setting and results.**

We train RFE-AD and UCB-AD for the *Car Driving* environment, with $4 \times 10^5$ training episodes and the parameters same as the Flappy Bird environment. We note that because the environment is $\mathcal{E}_1$, UCB-AD is trained with the knowledge of the distribution of the cell's type. Due to the large state space of this environment, it makes any planning algorithm computationally intensive. Thus we update and evaluate the policy for RFE-AD and UCB-AD every 8000 episode. The algorithms' parameters are the same as Flappy Bird environment. All experiments are repeated 5 times with the mean of results being reported.

As demonstrated in Figure 7b, UCB-AD not only outperforms RFE-AD but also achieves a near-optimal policy at a very early stage. We believe the strong performance of UCB-AD comes from the monotone property of Proposition 2 and that the optimistic policy happens to be the optimal policy in this setting. In Figure 7c, we further evaluate the RFE-$\beta$ algorithm for different $\beta$'s based on the estimated transition kernel from 7b for $\beta = 0.1, \ldots, 1$, and find $\{V_\beta^{\hat{\pi}_\beta}\}_{\beta > 0}$ close to $\{V_\beta^*\}_{\beta > 0}$ in a consistent manner.

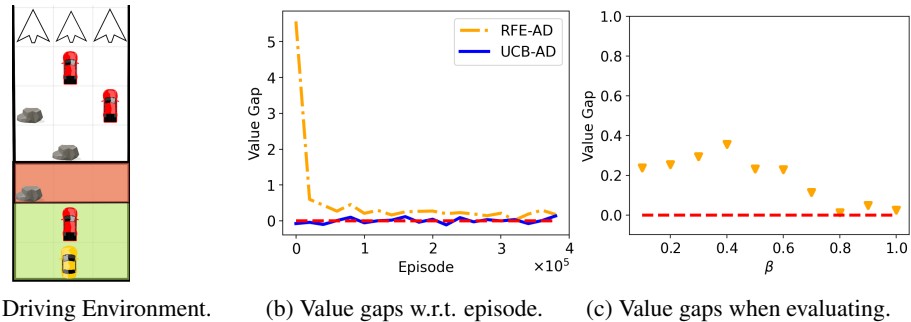

(a) Car Driving Environment.    (b) Value gaps w.r.t. episode.    (c) Value gaps when evaluating.

Figure 7: Environment and value gaps of algorithms. Car Driving environment (a), the driver needs to move forward as far as possible. The green cells are observable to the driver and the red cells (with the green cells) are observable to the machine. For (b) and (c), the value gap is defined as the difference between optimal value $V^*$ (or $V_\beta^*$) and $V^{\hat\pi}$ (or $V_\beta^{\hat\pi_\beta}$), with the red dashed line as the benchmark for $0$ loss of the policy. (b) shows the value gaps of RFE-AD and UCB-AD with respect to the training episode. (c) shows the value gaps when evaluating RFE-$\beta$ with $\beta = \{0.1, 0.2, ..., 1\}$.