# OpenReview forum: "Learning to Make Adherence-aware Advice"
_ICLR.cc/2024/Conference — ICLR 2024 poster_

### Official Review · Reviewer_tRbR · 2023-11-02

**Soundness:** 3 good
**Presentation:** 1 poor
**Contribution:** 2 fair
**Rating:** 5
**Confidence:** 3

**Summary:**

This paper studies the problem of learning to give advice to humans assuming humans may take the advice or not according to some underlying adherence level. The paper presents a formal problem definition of the problem, two UCB-style algorithms, and a theoretical analysis with convergence bounds.

**Strengths:**

1. **Problem formulation:** This paper proposes a new formulation of advising by assuming a fixed human player with a fixed probability of taking advice. This problem formulation to me is novel by considering the human adherence level.

2. **Theoretical analysis:** The paper adopts a few theoretical analysis frameworks to this advising problem and derives much-improved convergence bounds by leveraging the problem structures.

3. **Empirical studies:** It is nice to see the algorithm really works on the flabby bird domain although it is simple.

**Weaknesses:**

My biggest complaint about this paper is its presentation. A few comments are listed below.

1. It is weird that there is no citation at all in the introduction section. The introduction part is also particularly short with the contribution statements even deferred after the related work section. I would strongly encourage the authors to expand the introduction with more detailed explanations and more intuitions.

2. I think the "**Theoretical reinforcement learning**" section should be expanded a bit, e.g., to be an independent background section. I think this part is the most relevant literature. It would be better if the paper could spend more texts explaining the connection and differences between the existing literature and the current work from a more technical perspective.

3. It is strange to me that the convergence bounds are introduced even before the algorithms. This makes it difficult for the readers to follow the insights.

4. It is so tough for me to fully follow section 4.2. Equation (5) looks like a magical number to me and I have no idea how it is derived and why such a structure helps improve the bound from reward-free learning. I'm not a theoretical person. Carefully checking every detail of the proof is out of my capacity, so it would be great for me if the paper could explain well the insights of why the structure helps improve the bound for an outsider.

5. It is stated in the experiment section that two testbeds are considered, including the Flabby Bird domain and the Car Driving domain. However, I cannot find the Car Driving results.

**Questions:**

I don't have any specific questions.

I think this paper should be ultimately evaluated according to the strength of its theoretical results, which is out of my expertise. My current rating is based on its presentation but I'm okay to raise the score if other reviewers would champion its theoretical contributions.

---

> ### Author Response · Authors · 2023-11-14
>
> We thank the reviewer for taking the time to read our paper and for all the comments. Below are our responses to the raised questions.
>
> Two testbeds of the Flabby Bird domain and the Car Driving domain:
> - Sorry for the confusion. We mentioned in line 2 on page 8 that we defer the second experiment on Car Driving to the end of the appendix.
>
> Expanding the "Theoretical reinforcement learning" section:
>
> - We thank the reviewer for the suggestion. We presented the connection with theoretical reinforcement learning in the section Main Result, where we explained two informal theorems and discussed the connection and difference between the existing literature in detail. We believe that delving into technical details in the literature review section would be a bit too early, as by that time we haven't yet introduced the model and setting, which are critical in understanding the technical differences.
>
> - Moreover, we'd like to point out that the contribution of this work is beyond the sample complexity bound from theoretical reinforcement learning. Rather, the primary intention for the paper is to develop a new human-AI model for the sequential decision-making context and to derive algorithmic and theoretical insights for this new problem. The main motivation for the theoretical part is that we want to showcase the structural insights of the model (we can learn the model effectively by leveraging the adherence level), and connections for related areas (reward-free exploration and provably solving constrained MDP).
>
> More intuition for Equation (5):
>
> - We are happy to provide more explanation about the intuition of Equation (5). For theoretical online Reinforcement Learning algorithms, they usually maintain an uncertainty measure for every $(s,a)$ (state-action) pair and behave optimistically based on the uncertainty sets. Then, on page 6, we state that "The algorithm iteratively minimizes an upper bound defined by (5) which measures the uncertainty of a state-action pair, and the upper bound shrinks as the number of visits for the state-action pair increases. The algorithm stops when the upper bound is less than a pre-specified threshold."
>
>  - Most theoretical RL algorithms include constants in the algorithm that may appear as "magic numbers". The rationale behind these numbers often originates from a series of mathematical inequalities that drive the derivation of the sample complexity/regret bounds.
>
> We thank the reviewer again for reading our work and for all the helpful suggestions. And we hope the clarifications help to address the reviewer's concerns/confusion. In the following week, we will get back timely for any follow-up questions.

---

> > ### Comment · Reviewer_tRbR · 2023-11-22
> > **Thanks for the explanation**
> >
> > Based on the current form of the paper, I have not yet been fully convinced to raise my score. I may need to gather more feedback from other reviewers in the discussion section.

---

### Official Review · Reviewer_X7E8 · 2023-11-05

**Soundness:** 3 good
**Presentation:** 2 fair
**Contribution:** 3 good
**Rating:** 6
**Confidence:** 3

**Summary:**

This paper studies a reinforcement learning model that takes into account the human’s adherence level and incorporates a defer option so that the machine can temporarily refrain from making advice. The authors provide novel algorithms based on the principle of optimism in the face of uncertainty, which can learn the optimal advice policy and make advice only at critical time stamps. The authors further present the theoretical guarantee of the proposed algorithms and show their empirical performance.

**Strengths:**

This work investigates a novel reinforcement learning model taking into account new realistic factors, including the human 's adherence level and the defer option. The corresponding algorithm design and the theoretical analysis are novel to reinforcement learning. The empirical studies also verify the algorithms' performance. In addition, the paper is well-structured, and the main idea of this work is easy to follow.

**Weaknesses:**

(1) The major concern about this work is that there is a gap between the result for $\mathcal{E}_1$ in Algorithm 1 and that for $\mathcal{E}_2$ in Algorithm 3. The dependence on $S$ is typically more important than $H$ in the reinforcement learning problem. As claimed by the authors, the sample complexity bound for Algorithm 3 is sharper in the dependence on S than for Algorithm 1. But $\mathcal{E}_2$ should be a harder problem than $\mathcal{E}_1$, and thus the sample complexity bound for $\mathcal{E}_1$ is expected to be better. Can the authors discuss why there is such a gap and whether it is possible to modify Algorithm 1 to have a better sample complexity bound matching the one for Algorithm 3 in terms of $S$?

(2) Can the authors point out which part of the supplement is for the theorem and the corresponding proof of Algorithm 3? It appears not explicitly presented in the paper, although the authors have shown its sample complexity result in the main text.

(3) Eq.(1) is not clearly explained, especially the third case in Eq.(1). Can the authors provide some more explanation on how the probability of the third case is obtained?

**Questions:**

Please see the sections above.

---

> ### Author Response · Authors · 2023-11-14
>
> We thank the reviewer for the comments. We refer to the general response above for clarification on the positioning of our paper, and we respond to the reviewer's questions in the following.
>
> - Regret bounds: In the informal version of Theorem 1 on page 4, we mention that $O(H^2S^2A/\epsilon^2)$ is a PAC (probably approximately correct) sample complexity bound, which is different from the sample complexity bound $O(H^3SA/\epsilon^2)$ of Algorithm 3 (in line 32 and page 4, for Algorithm 3 we use sample complexity instead of PAC sample complexity. This type of sample complexity without PAC is also known as the sample complexity for BPI (best policy identification)). For more details of how PAC sample complexity and sample complexity differs, we refer to the statement of Theorem 1 and Theorem 2.
>
> -- In fact, to have a more comparable result to the sample complexity of $O(H^3SA/\epsilon^2)$ in Algorithm 3, we can just apply UCB-VI (Azar et al., 2015) to the machine's MDP (introduced in page 2), by the well-known trick that turns regret bound into sample complexity for BPI, and one will get the same sample complexity bound of $O(H^3SA/\epsilon^2)$ as in Algorithm 3. Notice that by simply doing this, there is no extra performance gain because neither Algorithm 3 nor this UCB-VI leverages the structure of the model.
>
> -- A natural question is why we mention Algorithm 3 for $\mathcal{E}_2$ instead of UCB-VI for $\mathcal{E}_2$ given that they have the same sample complexity. This is because Algorithm 3 comes from Algorithm 2, which aims to solve the reward-free exploration (RFE) problem instead of BPI. Therefore, Algorithm 3 is a more well-rounded algorithm and it can also perform RFE given more episodes (see Theorem 2), while UCB-VI is not designed for RFE.
>
> -- We believe that by leveraging the inherent property of our human-AI interaction model, one can tweak the parameters of Algorithm 1 and have a $O(H^2SA/\epsilon^2)$ sample complexity bound for BPI, and that would be more efficient than $O(H^3SA/\epsilon^2)$ of Algorithm 3 and UCB-VI. However, we choose to derive a better PAC sample complexity bound because the PAC sample complexity bound is a stronger result than the sample complexity bound for BPI.
>
> - The proof of Algorithm 3. We apologize for the caused confusion. The proof of Algorithm 3 starts from the last paragraph of page 18. We prove key lemmas for Algorithm 3 and use them to show the similar bound in Algorithm 2. This is because the only difference between Algorithm 2 and Algorithm 3 is that the terminating threshold for Algorithm 2 is $\epsilon/(2H)$ and for Algorithm 3 is $\epsilon/2$. The sample complexity for Algorithm 3 is $O(H^3SA/\epsilon^2)$, and for Algorithm 2 is $O(H^5SA/\epsilon^2)$. The extra $H^2$ comes from changing $\epsilon/2$ by $(\epsilon/2H)$ (which is also mentioned in lines 7 and 8 on page 20).
>
> - More explanations of Eq. (1). We gave some interpretation of Equation (1) on page 2 (above Definition 1). The first equation in (1) comes from our statement that if the machine chooses to defer, the human follows the default policy $\pi^\text{H}$. The second equation in (1) comes from our statement that If the machine chooses to advise, the human takes the machine’s advice with probability $\theta(s,a^{\text{M}})$. The third equation comes from the fact that if the human agent chooses not to adhere to the machine's advice (this happens with probability $1-\theta(s,a^{\text{M}})$, which is defined as the adherence level), the human will follow its fixed human policy $\pi_h^{\text{H}}(\cdot|s)$. However, because $a^{\text{M}}$ is already suggested by the machine and the human agent chooses not to adhere, the resulting policy will have support on $\mathcal{A}/\{a^{\text{M}}\}$ with distribution $\pi^{H}_h(\cdot|s)/(1-\pi_h^{\text{H}}(a^{\text{M}}|s))$ (we divide by $(1-\pi_h^{\text{H}}(a^{\text{M}}|s))$ to normalize the distribution).
>
> Lastly, we'd like to thank the reviewer again for all the questions raised by the reviewer. We will clarify these aspects in future versions of our paper. In the following days, we look forward to follow-up discussions, and we will get back to further questions timely.

---

> > ### Comment · Reviewer_X7E8 · 2023-12-01
> >
> > Thank the authors for the response. After reading the rebuttal, I think that my concerns are partially resolved. I would like to raise my score to 6. But I also think the presentation of this work needs to be improved in the future.

---

### Official Review · Reviewer_7r6M · 2023-11-11

**Soundness:** 2 fair
**Presentation:** 3 good
**Contribution:** 2 fair
**Rating:** 5
**Confidence:** 3

**Summary:**

This paper formalizes the setting of a static policy taking advice from another policy as an MDP. The authors use this abstraction to study the setting of a human taking advice from a machine in order to perform better on a decision-making task. Crucially, this MDP assumes the human (i.e. the static policy that receives machine advice) only follows the advice (i.e. a recommended action) with a probability $\theta$ per state. The paper then 1. introduces UCB-based algorithms to solve the case where the environment dynamics and human policy are know, but $\theta$ is unknown, and 2. adapts reward-free exploration methods to solve the case where the dynamics and advice-taking policy are additionally unknown, whereby the MDP is modeled as a CMDP. The experiments focus on a toy environment based on Flappy Bird and two toy, hard-coded policies ("Greedy" and "Fixed"). In this latter setting, the authors show their reward-free expoloration method, RFE-$\beta$ can successfully provide advice to improve the performance of the two fixed policies.

**Strengths:**

- This paper deals with the important topic of human-AI collaboration, a topic that is largely overlooked by the greater ML community in favor of fully-autonomous approaches. In particular, advice-taking is an important setting of human-AI collaboration, and the formalization of this setting as a CMDP, while straightforward, presents an important step toward making progress on this important problem.
- Overall, the presentation is highly legible and the key ideas are explain clearly.

**Weaknesses:**

While this paper studies an important topic, the paper should be improved before acceptance to a top-tier conference like ICLR:
- The experimental setting is extremely simplistic. The Flappy Bird MDP effectively consists of just 2 actions (up or down). The exact layout of MDP also appears fixed. Likewise, the advice-taking policies are fixed as two hard-coded policies. Effectively, the problem then reduces to learning a policy to solve 2 static MDPs with very small action spaces. Ideally the study can look at Flappy Bird under a procedurally-generated setting as well as look at other environments with higher-dimensional observation and action spaces.
- It would also be ideal to include an environment with continuous action spaces.
- The experimental setting also buckets all states into 2 coarse groupings of adherence levels for the advice-taking policy. In practice, the adherence levels are likely more complex and also time-varying. It would improve the paper to include experiments with a co-adapting advice-taking policy.
- In extending this work to higher-dimensional environments, the authors should consider comparing their approaches to existing deep RL methods for exploration [1,2] as baselines.

**Minor comments**
- In the related works section, the authors should include citations to the line of works studying human-AI cooperation in Hanabi:
    - Bard, Nolan, et al. "The hanabi challenge: A new frontier for ai research." Artificial Intelligence 280 (2020): 103216.
    - Foerster, Jakob, et al. "Bayesian action decoder for deep multi-agent reinforcement learning." International Conference on Machine Learning. PMLR, 2019.
    - Hu, Hengyuan, et al. "Off-belief learning." International Conference on Machine Learning. PMLR, 2021.


**References**

[1] Ciosek, Kamil, et al. "Better exploration with optimistic actor critic." Advances in Neural Information Processing Systems 32 (2019).
[2] Osband, Ian, et al. "Deep Exploration via Randomized Value Functions." J. Mach. Learn. Res. 20.124 (2019): 1-62.

**Questions:**

- Does this method handle higher-dimensional environments or continuous action spaces?
- How does this method handle more complex adherence functions?
- How does this method handle co-adapting advice takers?

---

> ### Author Response · Authors · 2023-11-14
>
> We thank the reviewer for spending time reading our paper, and for all the comments.
>
> We understand that this might be a last-minute/urgent review assignment for the reviewer, so we do appreciate all the comments and suggested references (we will include these in the future version of our paper). However, we believe the evaluation/criticism of a work on machine learning/deep learning should not be solely based on the experiments, but also on a large scope covering the modeling and theoretical contributions. To this end, we have created a TL;DR version of our positioning and contribution as in the general response. And we would really appreciate it if the reviewer could spend a bit more of time on reading these corresponding parts and evaluating these aspects of our paper, and in the following week, we are happy to address any follow-up comments/questions.
>
> In the following, we address the raised questions by the reviewer:
>
> "Does this method handle higher-dimensional environments or continuous action spaces?"
>
> - First, we note that we are presenting a model, not only a method. The model aims to provide a theoretical foundation so that it will be compatible with different extensions. To incorporate higher-dimensional environments, our model is compatible with the existing theoretical analyses of learning methods involving linear function approximation of the state space which provide a solution for high dimension environments. For continuous action space, we have to develop a continuous model separately. Just like the case in generic RL, the related theoretical property for continuous action is similar to the discrete one but it is out of the current scope of this work.
>
> "How does this method handle more complex adherence functions?"
>
> - We first note that the proposed model can be naturally related to other theoretical extensions to solve the aforementioned problem. If the adherence is nonstationary, the problem can be solved by learning the adherence level at every time step, which is a standard extension for RL algorithms. If the adherence level changes from episode to episode, we can rely on a standard approach by introducing a variational budget (Besbes, et. al. 2014) for the adherence level over episodes and get the corresponding theoretical result. If the variation of human adherence level exceeds the variational budget, the corresponding decision-making problem remains an open problem for the general theoretical RL/online learning community.
>
> "How does this method handle co-adapting advice takers?"
>
> - As we understand, co-adapting means that through interaction, the human will adapt to the machine and change its behavior policy or adherence level. This corresponds to a learning problem where the underlying MDP is changing, and if we are not controlling the amount of change in the behavior policy/adherence level, there is no model/algorithm that can provide theoretical guarantees for the co-adapting setting. We understand that some data-driven methods without theoretical guarantees can be very effective and practical for this scenario. We agree with the reviewer that this is an important future direction, and we will discuss this point more in the future version of our paper.
>
>
> Models and high-dimensional extensions: We appreciate the suggestions on extending the result to higher dimensions. We will mention this as a potential direction for future work.
>
> Empirical Experiment vs Numerical Experiment: We agree that experiments in a more practical setting are important. However, these experiments sometimes cannot be explained by any model or theory. In this work, if we conduct empirical experiments as suggested, it will be detached from our model because these scenarios are not related to our model at all. Instead, simpler experiments can either corroborate the underlying theory or be explained by it, making our work more sound. Furthermore, to ensure a more concise presentation of our paper, it is sensible to conduct experiments that are more closely related to our theoretical framework.
>
> We thank the reviewer again for the time spent reading our paper. We look forward to follow-up discussions.

---

### Author Response · Authors · 2023-11-14
**positioning and contribution**

We thank all the reviewers for spending time reading our paper, and for all the comments. We'd like to use this general comment to clarify the positioning and the contribution of our paper. Specifically, our paper aims to emphasize the following two aspects: (i) the introduction of a human-AI interaction model for the sequential decision-making context and (ii) a full theoretical analysis of the proposed algorithms that are also substantiated by numerical experiments.

- For (i), while there is a huge body of literature studying human-AI interaction for supervised learning (known as selective regression, learning with rejection, or learning to defer), the study of human-AI interaction for sequential decision-making is relatively in a lack. Our work provides some preliminary efforts in this pursuit.

- For (ii), there is a huge body of literature on theoretical/regret analysis of all kinds of RL algorithms. Comparatively, our theoretical analysis emphasizes (a) that by leveraging the problem structure, better regret performance is attainable theoretically and empirically; and (b) that the problem setup draws a natural connection with other frameworks such as reward-free learning and constrained MDP. Specifically, our model and analysis are compatible with, yet distinct from, the theoretical/regret analyses in the literature. For example, applying problem-agnostic algorithms and the associated theoretical analyses blindly to our model results in suboptimal performance, theoretically and numerically. Instead, the designed algorithms (such as Algorithm 1 and Algorithm 2) that are customized to the problem setup leverage the model's structure and integrate these analytical tools into these algorithms can provide better theoretical and empirical performance.

---

### Meta-Review · Area_Chair_mfEL · 2023-12-06

**Metareview:**

This paper introduces a sequential decision model for the setting where a system can suggest actions (i.e. provide advice) to an actor (e.g. a human) who has a certain adherence level (i.e. probability of following the advice). The authors provide theoretical analysis of this decision model, showing connections to reward-free learning and constrained MDPs. The authors then present algorithms directly exploiting the specific problem structure of this new decision model and show on an environment based on Flappy Bird that these bespoke algorithms lead to improved performance in terms of regret.

**Justification For Why Not Higher Score:**

While the theoretical contributions of this paper seem useful, as evidenced by the empirical results of algorithms exploiting this theory, the experiment setting is limited to a very toy setting. Thus the broader applicability and usefulness of the theoretical analysis that is the focus of this work is unknown.

**Justification For Why Not Lower Score:**

While the paper does not present compelling empirical results, the theoretical contribution here may be useful for others to build upon in what is an important decision making setting.

---

### Decision · Program_Chairs · 2024-01-16

Accept (poster)